# An intelligent method for Buoy meteorological data restoration using a Spatio-Temporal Dual-Attention Network with transformer and GAT

**Miaomiao Song**[1,2], **Jiuzhang Huang**[1], **Shizhe Chen**[2], **Xiao Fu**[1,2]*, **Shixuan Liu**[1,2], **Wenqing Li**[1,2], **Keke Zhang**[1,2], **Wei Hu**[1], **Xingkui Yan**[1,2]

**1** State Key Laboratory of Physical Oceanography, Institute of Oceanographic Instrumentation, Qilu University of Technology (Shandong Academy of Sciences), Qingdao, China, **2** Laoshan Laboratory, Qingdao, China

* dawn_fu@163.com

## Abstract

Meteorological sensors deployed on ocean buoys frequently suffer from data loss or outliers due to electromagnetic interference and component failures caused by harsh weather and environmental conditions. Accurate reconstruction of corrupted buoy data remains a significant challenge, as conventional interpolation and imputation methods often fail to capture the inherent spatio-temporal dependencies in marine meteorological variables. To address this issue, this paper proposes a novel deep learning model that integrates Transformer and Graph Attention Network (GAT) archi-tectures—termed the Spatio-Temporal Dual-Attention Network (ST-DAN). The model uses parallel computing to capture two aspects of the data: on one hand, it captures temporal dependencies through a Transformer enhanced by position encoding; on the other, it models inter-variable spatial correlations with a Graph Attention Network (GAT) based on a physically informed adjacency matrix, which dynamically adjusts the influence weights between variables to significantly enhance reconstruction accuracy. To evaluate the ST-DAN model, extensive experiments were conducted leveraging the ERA5 reanalysis dataset and in-situ observations from a Qingdao buoy, focusing on the reconstruction of temperature and wind speed data. The exper-iment result shows that ST-DAN outperformed baseline models (e.g., ARIMA, RNN, Bi-LSTM, and Transformer) across metrics including MAE, MSE, RMSE, and R². It indicates that the proposed model (ST-DAN) is off high robustness and achieves high-precision interpolation and anomaly correction for meteorological data.

## 1. Introduction

Meteorological research relies on long-term observational data to analyze regional climate characteristic. As crucial platform for acquiring meteorological and hydro-logical data, ocean buoys offer the advantages of long-term, continuous, and stable

**Data availability statement:** The complete code of this study has been publicly released on GitHub, including all datasets used in the experiment. The repository link is https://github.com/Khalil-gua/ST-DAN.git. This repository contains all relevant scripts and documents for viewing and copying.

**Funding:** This work is supported by Laoshan Laboratory Independent Innovation Science and Technology Project (Grant No. LSKJ202502405), the Taishan Industrial Program "Marine Observation and Detection Buoy Equipment R&D and Industrialization Team", Major Scientific Research Project for the Construction of State Key Laboratory at Qilu University of Technology (Shandong Academy of Sciences) (Grant No. 2025ZDGZ01) and Qilu University of Technology (Shandong Academy of Sciences) Major innovation project of science, education and production integration pilot project (2025ZDZX05). The funders had no role in study design, data collection and analysis, decision to publish, or preparation of the manuscript.

**Competing interests:** The authors have declared that no competing interests exist.

monitoring. They can consistently observe various parameters such as sea surface temperature, wind speed, wind direction, and atmospheric pressure. Real-time buoy data are widely used to test and refine Numerical Weather Prediction (*NWP*) models [1,2], which is of great significance for studying air sea interactions [3]. Multiple meteorological agencies, including the China Meteorological Administration and the European Centre for Medium-Range Weather Forecasts, utilize buoy data to enhance forecast accuracy; For instance, the *TRITON* buoy network monitors variation in the Pacific and Indian Oceans to study El Niño-Southern Oscillation (*ENSO*), monsoon, and interdecadal climate variability [4]. The Hong Kong Observatory has been experimenting with integrating buoy data into the machine learning models (*UWIN-CM*) to achieve real-time forecasting of tropical cyclones since 2024, thereby strengthening capabilities for predicting and preventing extreme weather events [5]. Furthermore, buoy data constitute a key component of climate change research, by providing reliable sea surface temperature observations [6], they contribute to improve the credibility of climate projections [7,8]. Consequently, the multi parameter data provided by buoys hold substantial value for enhancing weather prediction accuracy, understanding climate change, and monitoring the marine environment, the quality of these data directly impacts the reliability of scientific research results [9].

The marine meteorological environment is highly dynamic and variable. Buoy-based meteorological sensors are frequently susceptible to data gaps and anomalies due to a combination of external and internal factors. External challenges include long-term exposure to sunlight leading to component aging, sensor damage caused by strong winds or extreme cold, while internal issues encompass accuracy degradation from a lack of regular calibration and power insufficiency due to battery performance decay. Missing values and outliers can distort data characteristics, alter statistical properties of datasets—such as expected values and higher-order moments—and consequently increase the difficulty of data analysis. Therefore, it is essential to accurately identify and reconstruct anomalous data to provide reliable observational data for marine scientific research.

Conventional approaches to handling anomalous meteorological data often rely on statistical imputation methods [10]. These range from simple univariate techniques, such as mean, median, or mode filling, to more sophisticated models like K-Nearest Neighbors (KNN) and ARIMA. While straightforward to implement, simple statistical methods often fail to capture the variability and correlations in meteorological data, and they generally adapt poorly to the characteristics of time series. Yozgatligil et al. [11] compared various time series imputation methods, noting that although simpler algorithms (e.g., SAA, NR, NRWC) and computationally intensive approaches (e.g., EM-MCMC) are effective to some extent, they often come with high computational costs. The ARIMA model is more suitable for non-stationary series with trends or seasonal components and has been applied to tasks such as missing data estimation in water quality monitoring and multi-scale change point detection [12]. However, its reliance on specific distributional assumptions limits its performance on data that lacks significant trends.

With advancements in artificial intelligence, deep learning has demonstrated considerable effectiveness in data imputation tasks [13]. Classic sequential models such as *RNN, GRU*, and *LSTM* have demonstrated promising performance across various applications [14–18]. Nonetheless, they are often plagued by issues including vanishing or exploding gradients, difficulty in capturing long-range dependencies, high computational complexity, and a tendency to overfit [19–21]. Studies indicate that *LSTM*, while mitigating the problem to some extent, may still experience gradient vanishing [22], and its gating mechanisms can induce gradient anomalies when saturated [23]. In contrast, the Transformer architecture, by leveraging a global self-attention mechanism, demonstrates a enhanced capability in capturing long-term dependencies and complex temporal patterns, thereby offering a more effective solution for imputation in long sequence data [24,25].

With the widespread application of Transformer in temporal prediction, various optimization variants such as Informer [26] and Auto-former [27] have been proposed to improve the computational efficiency and expressive power of the original model. These models have been proven effective in multiple experiments through sparse attention or sequence decomposition mechanisms [27–32]. However, meteorological data has both spatiotemporal dependencies, and modeling only temporal features can easily ignore spatial correlations, and vice versa [13].

Unlike structured data such as images, spatial dependencies in meteorological data often exists as irregular graph structure (non-Euclidean data), which necessitate processing with Graph Nural Network (*GNN*). Representative GNN architectures include Graph Convolutional Networks (*GCN*) [33,34], Graph Attention Network (*GAT*) [35] and Graph Auto-encoders encoder (*GAE*) [36]. These models effectively capture complex spatial dependencies through graph convolution operations, attention weighted aggregation and autoencoding mechanisms, respectively, making them suitable for tasks such as node classification and link prediction.

To enhance the physical consistency and accuracy of meteorological data repair, this paper constructs an advanced spatio-temporal joint modeling framework, termed the Spatio-Temporal Dual Attention Network(*ST-DAN*), The *ST-DAN* model effectively integrates Transformer encoder, physical constrained matrix and a Graph Attention Network(*GAT*) to form a dual link reasoning architecture. This design enables the separate extraction and subsequent fusion of temporal and spatial features for predictive repair. In the temporal encoding link, the Transformer encoder structure is used to capture the long-distance features of time series; In the spatial encoding link, the *GAT* model extracts inter-variable spatial dependence. Crucially, the native attention mechanism in GAT is replaced by a physically constrained adjacency matrix, which effectively suppresses connections between irrelevant features while strengthening correlations between related ones. This matrix dynamically adjusts influence weight during training, thereby enhancing prediction accuracy while reducing computational overhead. In general, the *ST-DAN* model successfully leverages both temporal and spatial constraints to achieve effective restoration of meteorological data.

The remainder of this paper is organized into five sections. The first section presents the research background and the motivation behind the proposed model; The second section focuses on the algorithm, structure, working principle and function of each part of the model; The third section includes the processing of the experimental data set and the arrangement of the experimental environment; The fourth section is consisted of experiments and analysis of the experimental results; The fifth section contains the conclusion and future prospects.

## 2. Methodologies

### 2.1. Modeling of buoy meteorological data restoration

In the modeling process of buoy meteorological data restoration, the input data
$X = [DateTime, feature1, feature2, feature3 \ldots]$ is first divided into meteorological features
$X_t^{weather} = [feature1, feature2, feature3, \ldots]$ and temporal features $X_t^{temporal} = [feature_{target}, day, hour, \ldots]$ including target features. These features are then fed into the spatial encoding link and the temporal encoding link respectively. In the spatial encoding link, we introduce a unique physical constraint matrix $M_{ij}$ to replace the *GAT* neural network with the attention

mechanism, which can avoid irrelevant features connections and output the spatial feature $S_{feature} = GAT(x_t^{weather}, M_{ij})$ after training and learning; in the temporal encoding link, the data is first encoded by position, then entered into the multi-layer transformer encoder, and output the temporal feature $T_{feature} = Transformer(PE(x_t^{temporal}))$, and $PE$ is the positional encoding. The features output from the two links are fused through the feature fusion layer to obtain the fusion feature $F_{fused}$.

In the prediction output stage, the sliding window technique is used to convert the original sequence into input–output sample pairs to effectively model temporal dependencies and feature interactions.. Given the temporal series $X = \{x_1, x_2, ..., x_N\} \in R^{T \times d}$ with length $N$, where $T$ is the length of the temporal series and $d$ is the feature dimension, training sample $(X_i, \hat{y}_i)$ are generated via the sliding window. Here the input sequence $X_i = \{x_i, x_{i+1}, ..., x_{i+L-1}\}$ contains L time step features, and the target value $\hat{y}_i = x_{i+L}$ corresponds to the target variable at the next time step. This approach enables the model to learn the mapping from the past $L$ time steps to the next $P$ time steps. For example, $n$ the dataset used in this experiment, where the temporal resolution of the data is 1 hour, we set $L=24$ and $P=1$. That is, when a data point is identified as an anomaly or a missing value, the 24 hours of data preceding that point are used to predict and replace its value, thereby achieving data repair. Additionally, to handle consecutive missing values and avoid error accumulation, a bidirectional prediction strategy is adopted at the prediction stage. This means that the input sequence is first predicted in the forward temporal direction and then in the reverse temporal direction. The final predicted output is obtained as follows:

$$\hat{y}_i = \frac{(1-\alpha)F_{fused}^{forward} + \alpha F_{fused}^{backward}}{2}$$

(1)

Where $\hat{y}_i$ represents the final predicted value, $F_{fused}^{forward}$ denotes the forward fusion feature, and $F_{fused}^{backward}$ denotes the backward fusion feature. To ensure the reasonable prediction output, dynamic weights is applied based on the relative position of the predicted point within the gap of missing data. A higher weight is assigned to positions closer to the prediction direction, with the weight constrained to the range [0.1, 0.9]. The weighting function is defined as follows: $\alpha = 0.1 + 0.8 \times \frac{position}{length-1}$.

## 2.2. Constructing spatio-temporal dual attention network model

The dual-path architecture, owing to its capability to collaboratively process features of different natures within data, has demonstrated significant technical advantages across numerous fields. Addressing the characteristics of meteorological data, which inherently encompass both their own temporal patterns and spatial correlations among elements, this study adopts a dual-path structure and proposes a Spatio-temporal Dual Attention Network (ST-DAN) that integrates Transformer and Graph Attention Networks (GAT), aiming to achieve high-precision time-series data imputation. The temporal path of this model employs Transformer encoder layers, specifically designed to capture the dynamic temporal features of individual meteorological elements [37]. The spatial path utilizes GAT to model the complex interrelationships among different meteorological elements [38]. Furthermore, this study introduces a key optimization to the attention mechanism by incorporating a novel physical adjacency matrix to replace the traditional attention weight generation mechanism. This enables the dynamic and appropriate adjustment of influence weights among different elements, allowing the model to capture dependency relationships that more closely align with real physical processes, thereby yielding more accurate prediction results.

The overall architecture of *ST-DAN* is shown in Fig 1, which is composed of a temporal encoding link, a spatial encoding link, and a features fusion layer. The temporal encoding link and the spatial encoding link process the original input data is in parallel, and the features fusion layer is performed to finally output the prediction data. Prior to data input, the *Z-score* method is used to standardize the data, transforming the input features into a distribution with a mean of 0 and a

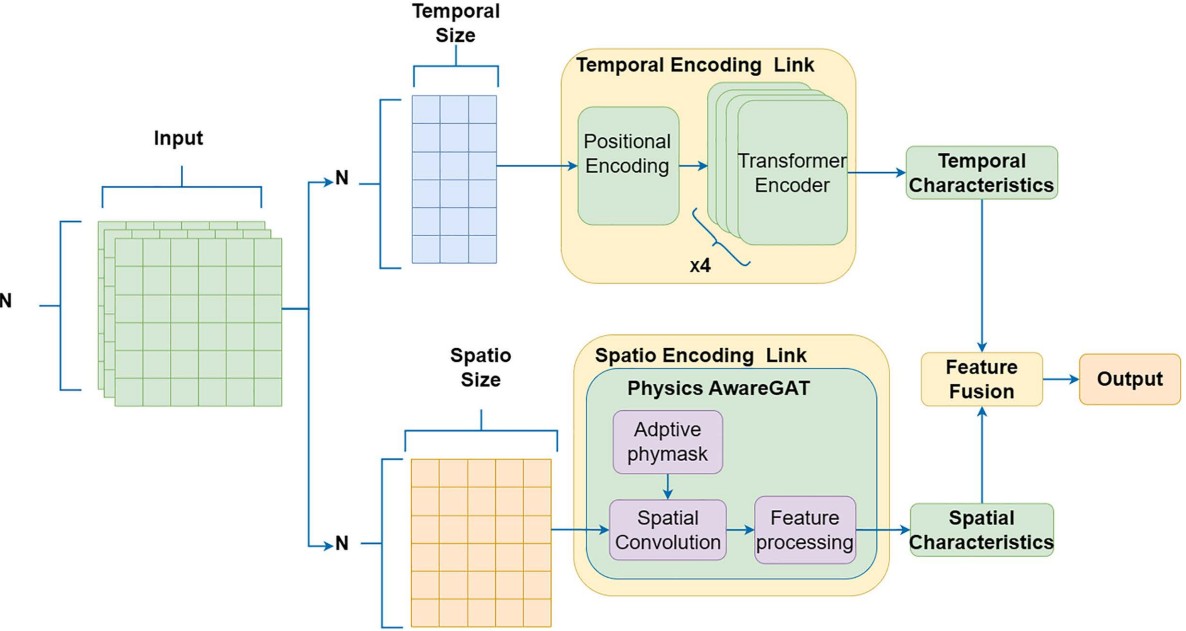

**Fig 1. Network architecture of the ST-DAN model.**

standard deviation of 1, so as to improve the stability of model training. The input data grid contains $m$ dimensional temporal elements, such as year, month, day, hour, minute, etc.; It also includes $n$ dimensional meteorological elements, such as temperature, pressure, wind speed, wave period, etc. The data are then split into m-dimensional temporal features plus 1-dimensional target meteorological elements, which enter the temporal encoding link as $x_t^{temporal}$; All n-dimensional meteorological elements enter the spatial encoding link as $x_t^{weather}$. The temporal encoder first encodes the observation time using a position encoding function, and then employs a multi-layer Transformer encoder, which uses its attention mechanism to capture the temporal dependence in the data and output temporal features; The spatial encoding link uses a graph attention network to learn the relationship among multiple variables and output spatial features. The temporal and spatial features are then fed into the feature fusion module. A decoupling fusion strategy is applied, in which a gating mechanism—generated via linear transformation and a sigmoid function—dynamically adjusts the weights of the temporal and spatial features. The module finally produces a fused representation, based on which the final prediction result is generated. The detailed algorithms and working principles of each module in the ST-DAN model are described as follows.

**2.2.1. Temporal encoding link.** In the temporal encoding link, a temporal encoder is constructed to fuse time information and target features. The temporal encoder adopts a multi-layer Transformer encoder structure, as illustrated in Fig 2. A four-layer Transformer encoder is adopted in this architecture. The self-attention mechanism contained in the Transformer encoder enables the model to capture long-range temporal dependencies in the data, thereby effectively fitting the evolving trends over time and obtaining representative temporal features.

After the input sequence $X_t^{temporal}$ enters the temporal encoding link, it will be projected the original $L*(m+1)$ dimension to $L*d_{model}$, $d_{model}$ is the Transformer encoder model dimension, which is set to 256 here. Then the position encoder is used to encode the temporal information and combined it with the target feature. Because the Transformer model only focuses on the content similarity between elements and lacks the ability to perceive the order of elements, positional encoding is essential. The positional encoder uses the sine/cosine function (as shown in formulas (2) and (3)) to generate a fixed position encoding vector [39].

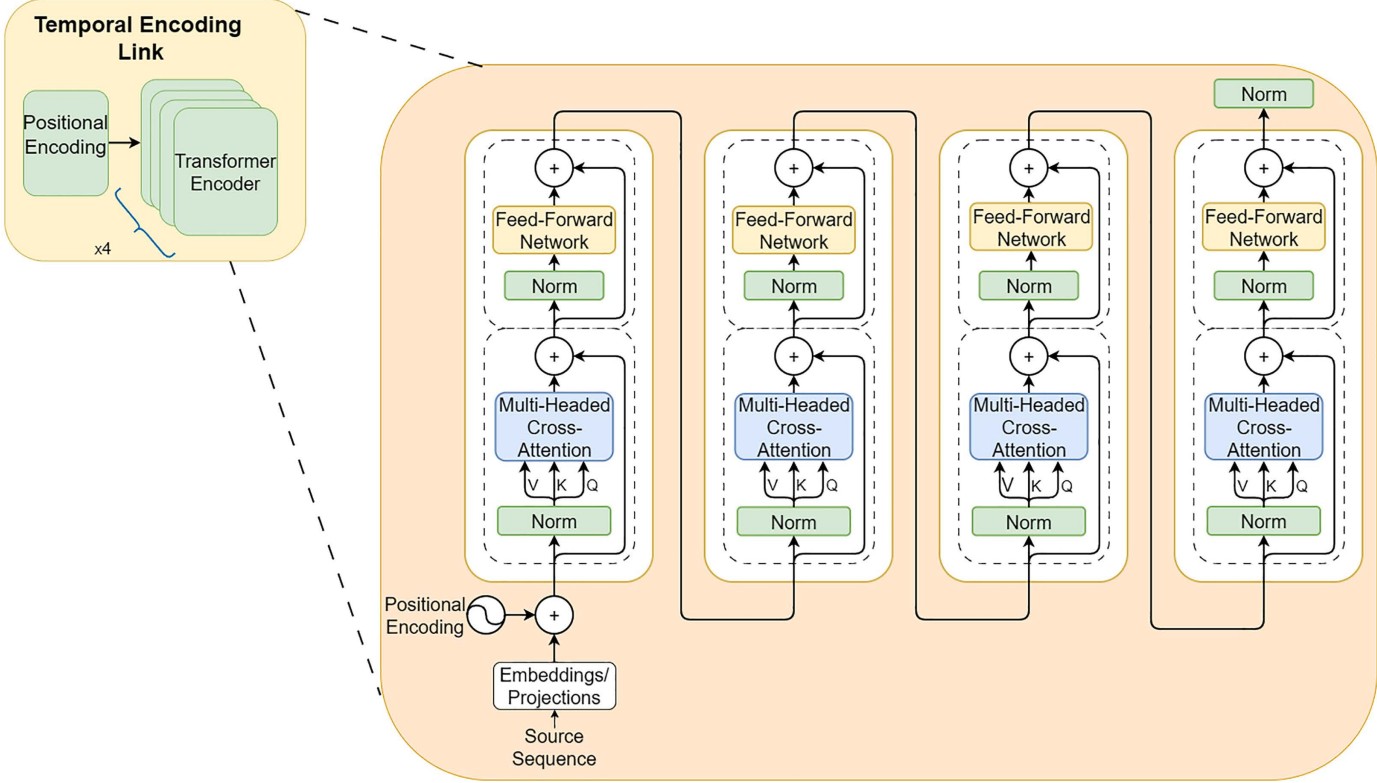

**Fig 2. Structure of the Transformer-Encoder-based Temporal reasoning component.**

$$PE_{(pos,2i)} = sin(\frac{pos}{10000^{2i/d_{model}}})$$

(2)

$$PE_{(pos,2i+1)} = cos(\frac{pos}{10000^{2i/d_{model}}})$$

(3)

Where *pos* is the time step position, *i* is the dimension index, $PE_{(pos,2i)}$ is the positional encoding of the even dimension index, and $PE_{(pos,2i+1)}$ is the positional coding of the odd dimension index. The position encoder generates a position matrix $PE_i = \{pe_i, pe_{i+1}, ..., pe_{i+L-1}\}$ based on the input data

The input sequence $X_{PE}^{temporal}$ after positional encoding fusion will be input into the attention mechanism layer, and its calculation logic is shown in Fig 3. First, $X_{PE}^{temporal}$ will be multiplied by the attention weight matrices $W^Q, W^K, W^V$ respectively to obtain $Q$, $K$, and $V$, which correspond to the query, key, and value matrices respectively. Then, the attention score will be calculated. The calculation method of the attention score is shown in formula (4).

$$Attention(Q, K, V) = softmax\left(\frac{QK^T}{\sqrt{D_{out}}}\right) V$$

(4)

Where *Attention(Q,K,V)* represents the attention score, $D_{in}$ denotes the dimension of the input data $X_{PE}^{temporal}$, and $D_{out}$ represents the dimensions of $W^Q, W^K, W^V$.

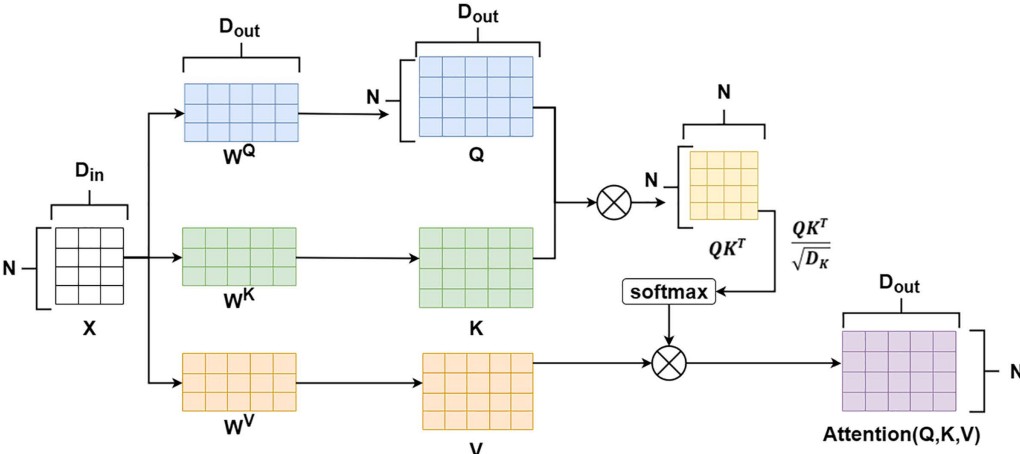

**Fig 3. Schematic diagram of the self-attention mechanism.**

The multi-head attention mechanism performs multiple attention operations on the same input. Each attention module, or "head," has the same structure but uses independently learned weight matrices, allowing the model to jointly attend to information from different representation subspaces at different positions. This enables the model to capture more extensive and diverse contextual features and enhances its expressive power [39].

After the attention score $Attention(Q, K, V)$ is computed, it is combined with the original input sequence $X_{PE}^{temporal}$. The calculation process is as follows:

$$x = LayerNorm(X_{PE}^{temporal} + Attention(Q, K, V))$$

(5)

$x$ represents the layer normalization result, and Layer Norm is the layer normalization function.

Subsequently, the representation $x$ enters the Feed-Forward Network (FFN). The FFN acts independently on the representation vectors at each position in the sequence output by the self-attention layer. Its main function is to perform nonlinear transformation and deep feature abstraction on each position vector that already contains global context information. Its structure follows an expansion-nonlinearity-contraction design. First, the input vector $x$ is linearly projected to a higher dimension $d_{ff}$ through a weight matrix $W_1 \in R^{d_{model} \times d_{ff}}$ and a bias $b_1 \in R^{d_{ff}}$, where $d_{ff} = 4 \times d_{model}$ and we set $d_{model}$ to 256. Then, a nonlinear activation function, specifically the $ReLU$ function, is applied to the projected result. Afterwards, the activated high-dimensional vector is linearly projected back to the original $d_{model}$ dimension through a weight matrix $W_2 \in R^{d_{model} \times d_{ff}}$ and a bias $b_2 \in R^{d_{ff}}$, according to the mathematical formula:

$$FFN(x) = ReLU(xW_1 + b_1)W_2 + b_2$$

(6)

Among them, The function $FFN(x)$ represents the abstraction of features after nonlinear transformation and computation through the feedforward fully connected layer. The parameters ($W_1$, $W_2$, $b_1$, $b_2$) of $FFN$ are shared across all sequence positions within the same layer, but the computation is position-independent.

$FFN$ complements the self-attention mechanism, which focuses on learning inter-position dependencies, and together they form the powerful feature learning capability of the Transformer encoder layer. By introducing nonlinear transformations and high-dimensional mappings, the $FFN$ layer significantly enhances the model's representational capacity

and serves as a key component for the model to learn complex features while contributing a substantial number of parameters.

Subsequently, $FFN(x)$ performs another residual connection and layer normalization with $x$, ultimately obtaining and outputting the temporal feature matrix $T_{feature} \in R^{B \times L \times d_h}$ of the target element. Here, $B$ represents the batch size, and $d_h$ denotes the size of the hidden layer. The calculation process is as follows:

$$T_{feature} = LayerNorm(x + FFN(x)) \tag{7}$$

**2.2.2. Spatial encoding link.** In the application of graph neural networks, modeling based on graph structures can effectively capture the complex relationships between variables. The introduction of attention mechanisms further enhances the model's ability to capture dynamic spatial dependencies [40]. Additionally, architectures that integrate graph convolution with temporal convolution have been shown to significantly improve the modeling performance of spatiotemporal sequence data, achieving superior results in prediction tasks involving multi-source heterogeneous data [41].

Inspired by this, the present study treats the sequences of various meteorological elements as nodes in a graph structure and designs a Physics-aware Spatial Encoder (PhysicsAwareGAT), whose core is an improved Graph Attention Network (Fig 4). This encoder uses the correlation coefficient graph between meteorological elements (Fig 5) as its underlying structure. Through the graph attention mechanism, it adaptively learns the strength of associations between elements, thereby achieving effective modeling of spatial correlations.

After the input sequence $X_t^{weather}$ enters the spatial encoding link, an adaptive physical constraint matrix is generated in the Spatial Encoding Link. The attention mechanism in traditional graph attention networks relies on data-driven implicit learning relationships, which can automatically capture the relationships between features and dynamically adjust weights. This is extremely convenient and effective for some data where the correlation between features cannot be directly known. However, during the learning process, there may be cases where false correlations are captured and lack physical interpretability. For meteorological data, we can directly know the relationships between features, such as temperature being related to wind speed and pressure, but not to precipitation. The adaptive physical mask matrix mechanism proposed in this invention replaces the traditional attention mechanism. Through a predefined physical constraint matrix, it can forcefully prohibit connections that violate domain knowledge, ensuring that elements with existing correlations can be correctly connected. This enhances the interpretability of the model while avoiding unnecessary calculations and improving model accuracy and training efficiency.

First, a physical prior matrix $Aij$ (an n*n Boolean matrix, as shown in Fig 6) is preset,. Its rows and columns correspond to the set of meteorological features $\{\alpha_1, \alpha_2, ..., \alpha_n\}$, where $n$ is determined by the number of meteorological features in the collected data. In this matrix, the row index represents the target feature (i.e., the affected feature), while the column index represents the source feature (i.e., the influencing feature). For each row corresponding to a target feature, the entries in the columns associated with the source features are set to either 0 or 1: 0 indicates that the source feature is irrelevant to the target feature, and 1 indicates that the source feature is relevant. The values of this matrix must be manually defined based on expert knowledge.

Using the example of $X_t^{weather}$ in this experiment, the five meteorological features $\{\alpha_1, \alpha_2, ..., \alpha_5\}$ correspond to temperature, wind speed, pressure, wave period, and humidity, respectively. If the goal is to predict temperature, the first row can be set to $[1, 1, 1, 0, 1]$; if predicting wind speed, the second row can be set to $[1, 1, 1, 0, 1]$, This approach effectively eliminates correlations between irrelevant features.

However, the Boolean matrix $Aij$ can only express the irrelevant or relevant relationships between meteorological features (i.e., values of 0 or 1), but it cannot quantify the influence weights between the elements. To address this, we introduce an adaptive physical constraint matrix module. This module remaps $Aij$ into a physical learning matrix $Mij$, whose

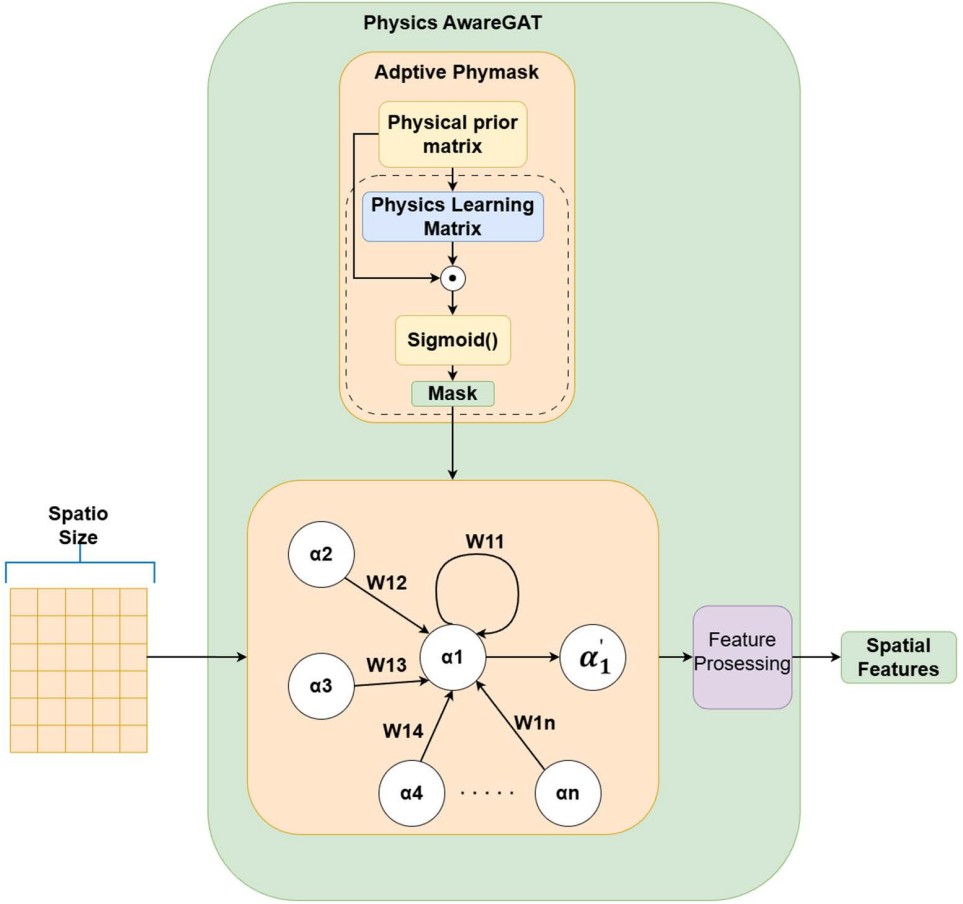

**Fig 4. Structure of the Physics Aware GAT.**

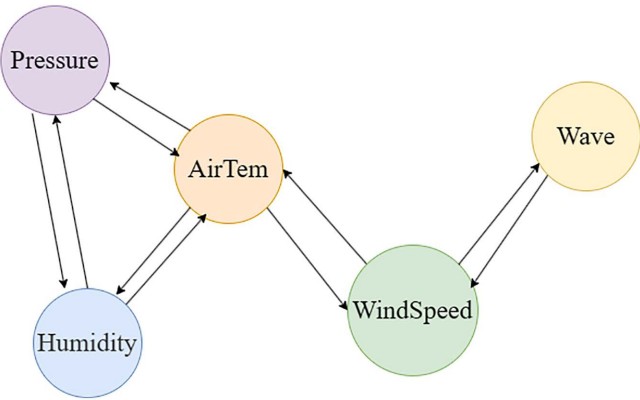

**Fig 5. Graph structure of meteorological element correlation.**

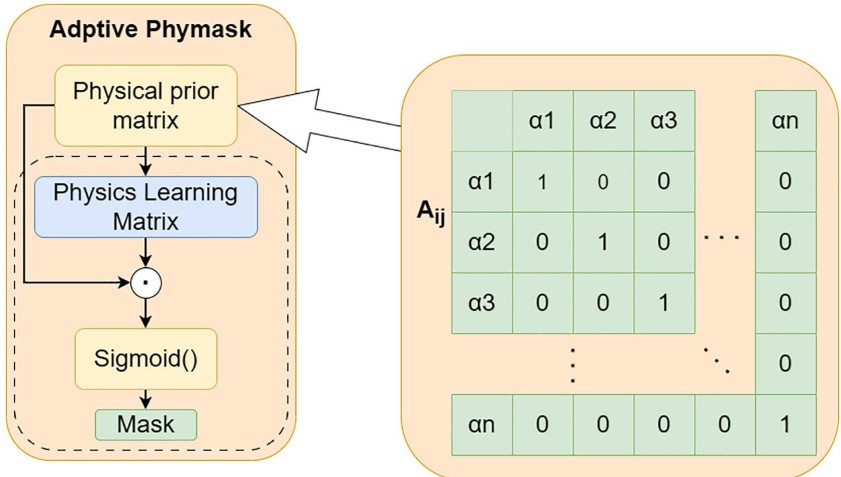

**Fig 6. Schematic diagram of adaptive physical constraint matrix.**

initial values are consistent with *Aij*, but the type is converted from a Boolean matrix to a floating-point matrix, serving as a learnable parameter matrix. To broaden the value range, increase the optimization gradient, and make weight changes smoother, the weight range of *Mij* is constrained to [0, 2]. This setting uses 1 as the center point: values greater than 1 enhance correlation, while values less than 1 weaken correlation. This interval amplification helps avoid the issue of universally small weights and enhances distinguishability.

In each training round, *Mij* functions similarly to a convolution kernel, sliding along the temporal dimension with a specified stride over the data, and performs feature extraction via matrix multiplication. After the training round, the parameters of *Mij* are updated using gradient descent. During this process, *Aij* remains fixed. Through the Hadamard product of *Mij* and *Aij*, the weights corresponding to originally irrelevant elements are guaranteed to remain zero, thereby forcibly masking irrelevant connections.

The updated weight matrix is normalized to the [0, 1] interval using the Sigmoid function, then multiplied by 2 to restore it to the [0, 2] range, generating a new matrix $M'_{ij}$ for the next training round. This mechanism effectively prevents weights from exceeding bounds due to cumulative effects during iterations. Finally, $M'_{ij}$ is output as the adaptive physical constraint matrix for subsequent prediction tasks. Its calculation formula is as follows:

$$M'_{ij} = \sigma \left( M_{ij} \odot A_{ij} \right) \times 2 \tag{8}$$

Where $\sigma$ denotes the Sigmoid function and $\odot$ denotes the Hadamard product. This process achieves an effective integration of physical priors and data-driven learning, retaining the physical constraints of feature relationships while allowing the model to adaptively adjust the influence weights. In the actual computation process, the weight range is constrained to [0,2], where 0 indicates irrelevance and 2 indicates strong correlation. After obtaining the adaptive physical constraint matrix $M'_{ij}$, $M'_{ij}$ is used as a mask to perform graph convolution operations with the input sequence $X_t^{weather}$. The graph convolution calculation is defined as follows:

$$\alpha'_i = \sum_{j \in N(i)} M'_{ij} \alpha_j \tag{9}$$

Where $\alpha_i^{'}$ represents the meteorological features obtained after calculation, $\alpha_j$ represents the original value of node $j$, and $N(i)$ represents the set of all values that are 1 in the $i$-$th$ row of $A_{ij}$. At this stage, the meteorological features are not yet the final spatial features and require be further transformation through the feature processing layer.

In the feature processing layer, the obtained meteorological features $\alpha_i^{'}$ undergo a nonlinear transformation via the $ReLU$ activation function and are projected into a high-dimensional space, ultimately yielding the spatial features $S_{feature} \in R^{B \times L \times d_h}$ of the target element as output. The calculation process is as follows:

$$S_{feature} = W_4 \left( ReLU \left( \alpha_i^{'} W_3 + b_3 \right) \right) + b_4$$

(10)

Where $W_3$ and $W_4$ are the weight matrices of spatial encoding link, and $b_3$ and $b_4$ are the bias vectors of spatial encoding link.

**2.2.3. Features fusion.** The application of gating mechanisms in deep learning models for feature fusion has become a key technology for enhancing the efficiency of multi-modal and multi-level information integration. This mechanism dynamically regulates the contribution of features from different sources or levels, enabling the selective retention of critical information and the suppression of noise, thereby significantly improving the model's representational capacity and generalization performance. It has been widely applied and proven effective in time series forecasting [42–44].

In this experiment, after the spatial and temporal pathways output spatial and temporal features respectively, a gating mechanism is similarly adopted for feature integration. The feature fusion module in this study achieves the organic integration of multi-modal information through a decoupling-fusion strategy. Its design adheres to the following principles: Physical Consistency, ensuring spatial features carry explicit meteorological physical relationships; Temporal Dynamics, capturing periodic and trend patterns within temporal features; and Adaptive Interaction, dynamically adjusting the contribution of spatio-temporal features via the gating mechanism. The structure diagram is shown below (Fig 7).

Among them, $W_5 \in R^{2d_h \times d_h}$ and $W_6 \in R^{d_h \times d_h}$ are the learnable weight matrices of the linear fusion layer. Spatial Features represents the feature matrix $S_{feature} \in R^{B \times L \times d_h}$ output by the improved graph attention network in the spatial link; Temporal features denotes the temporal feature matrix $T_{feature} \in R^{B \times L \times d_h}$ generated by the Transformer encoder after concatenating the target meteorological features and temporal features in the temporal link. $Weight\_T$ and $Weight\_S$ represent the weights assigned to temporal features and spatial features, respectively. After concatenating them in the hidden layer dimension, adaptive weight allocation is achieved through a gated fusion network: first, a gated vector is generated via linear transformation and Sigmoid activation

$$g = \sigma \left( W_6 \cdot \sigma \left( W_5 \cdot Concat \left( S_{feature}, T_{feature} \right) \right) \right)$$

(11)

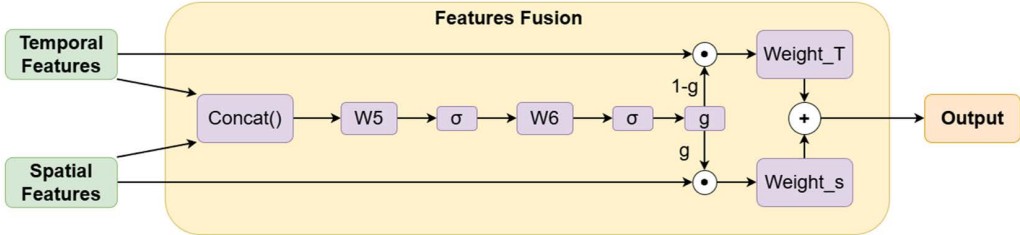

**Fig 7. Architecture of the features fusion layer.**

Then through:

$$F_{fused} = g \odot S_{feature} + (1-g) \odot T_{feature} \tag{12}$$

To achieve feature fusion, this design ensures the physical consistency of spatial features through a physical mask matrix, while utilizing a dynamic weight matrix to adjust the contribution of spatio-temporal features.

After the model training is completed, during the application phase, the data is fed into the trained model, and the last time step of the output is taken as the predicted value to replace the outliers, thereby accomplishing data repair.

**2.2.4. Loss and evaluation metric.** The loss function adopts Mean Absolute Error (*MAE*) as the training objective, which is calculated as shown in formula (13). Here, $M$ represents the number of samples, $y_i$ denotes the true value, and $\hat{y}_i$ represents the predicted value. *MAE* exhibits robustness against physical outliers. When faced with extreme values in meteorological data, *MAE* is less affected, meeting the fault tolerance requirements of meteorological prediction regarding outliers. Additionally, *MAE* has the advantage of interpretability, with its physical units consistent with the prediction target, facilitating direct evaluation of model performance. During the training process, a gradient clipping mechanism is introduced to prevent gradient explosion. The calculation of gradient clipping is shown in formula (14). Here, $\theta$ is the gradient threshold, preset to 1, $g$ is the current gradient, and $\|g\|_2$ is the *L2* norm of $g$.

$$LOSS = MAE = \frac{1}{M}\sum_{i=1}^{M}|y_i - \hat{y}_i| \tag{13}$$

$$g' = g \cdot min(1, \frac{\theta}{\|g\|_2}) \tag{14}$$

In addition to using MAE as the loss function, we also employed MSE, RMSE, R², and AR² as evaluation metrics. The calculation formulas are as shown in Formulas 15–17. MSE stands for Mean Squared Error, RMSE for Root Mean Squared Error, R² for the Coefficient of Determination, and AR² for the Adjusted Coefficient of Determination. All five metrics can reflect the discrepancy between the predicted data and the true values. The four metrics generally range between 0 and 1, with MAE, MSE, and RMSE being more effective when closer to 0, while R² and AR² are more effective when closer to 1. Here, n represents the sample size, and k denotes the number of explanatory variables.

$$MSE = \frac{1}{m}\sum_{i=1}^{m}(y_i - \hat{y}_i)^2 \tag{15}$$

$$RMSE = \sqrt{\frac{1}{m}\sum_{i=1}^{m}(y_i - \hat{y}_i)^2} \tag{16}$$

$$R2 = 1 - \frac{\sum_i(\hat{y}_i - y_i)^2}{\sum_i(\bar{y}_i - y_i)^2} \tag{17}$$

$$AR2 = 1 - [\frac{(1-R^2)(n-1)}{n-k-1}] \tag{18}$$

## 3. Experimental data processing and hyperparameter setting

### 3.1. Experimental data

In the experiment, the ERA5 dataset was first used for experimental verification. The selected area was Qingdao, China, with latitude 36.00−36.25N and longitude 120.25−120.50E. The time span was three years from January 1, 2022, to December 31, 2024, with a temporal resolution of one hour and a total of 26,304 records. The observed elements included temperature, pressure, wave period, wind speed, and rainfall. Training and prediction were conducted with temperature as the target feature. Then, buoy data measured in the Xiaomai Island sea area of Qingdao was used for testing. A total of 42,646 data records with a temporal resolution of ten minutes from June 1, 2024, to June 1, 2025, were selected. The measured data includes five elements: temperature, wind speed, pressure, wave period, and humidity. Training and prediction were conducted for wind speed. During the experiment, the data was divided into a training set $X = \{x_1, x_2, \ldots, x_n\}$ and a test set – true value $Y = \{y_1, y_2, \ldots, y_n\}$ at a ratio of 8:2. Then, 20% of the data in the test set – true value was modified as outliers for model prediction. Missing values and outliers were randomly added to the test set. First, 15% of the temperature data in the test set – true value was randomly modified as missing values. Considering that missing values may occur consecutively in actual situations, the maximum consecutive number of missing values was limited to 4 to mimic missing values in actual collected data. These data were then marked to obtain $Y_{Nan}$. Subsequently, 5% of the data was modified as Gaussian noise values.

### 3.2. Data noise injection

We have designed an adaptive Gaussian noise injection method based on stratified sampling. The noise injection process comprises the following three steps.

The first step involves randomly sorting the candidate set $V$, where $V$ represents the non-null values in the data $Y_{Nan}$. Subsequently, $V$ is divided into $D$ layers, denoted as $V = \{V_1, V_2, \ldots, V_D\}$. The number of outliers $m$ to be allocated to each layer is calculated using the following formula.

$$m = \frac{Q}{D}$$

(19)

Where $Q$ represents the total number of noise points to be inserted. For the remainder $R = Q \bmod D$, allocate one additional sample to each of the first $R$ layers to ensure as balanced a sample distribution as possible. Once the number of outliers for each layer is determined, independently draw $m$ original data points $x_i$ from each layer.

The second step involves generating adaptive Gaussian noise. The intensity of noise is crucial to the model's prediction performance. In this study, Gaussian noise is adaptively generated based on the standard deviation of the original data. The noise generation function is $\in \sim \mathcal{N}(0, (\mu \cdot 0.5)^2)$, where $\mu$ represents the standard deviation of the original data. By scaling the standard deviation by 0.5, the noise intensity is controlled within a reasonable range, ensuring that the noise can introduce data variations without excessively distorting the original data characteristics.

The third step injects the generated noise $\in_i$ into $y_i$ to obtain the data point $y_i'$ after noise injection. The calculation formula is:

$$y_i' = y_i + \in_i$$

(20)

Through index matching, this process precisely adds noise to selected data points, injecting a specific proportion of noise uniformly across the time dimension. This approach enables better simulation of real-world scenarios while preserving the distribution characteristics of the original data. After following the aforementioned steps, a test set $Y$ containing 15% missing values and 5% outliers is obtained.

### 3.3. Experimental environment hyperparameter settings

The hardware configuration of the experimental environment used in this study includes an *NVIDIA GTX 1050ti GPU*, an *Intel Core i5-11400F CPU,* and *32GB DDR4* memory. The software configuration includes the deep learning framework *Pytorch 2.7*, the deep learning computing component *CUDA 12.4*, the data processing libraries *pandas 2.2.3* and *scikit-learn 1.6.1*, and the visualization component *Matplotlib 3.9.2.*

For the *ERA5* dataset, the adopted training hyperparameters are shown in Table 1, where the input length and prediction length are represented as 24 data points, i.e., predicting one data point for 24 consecutive hours. To ensure that the model can correctly capture the dependencies in the data while also considering computational efficiency, after multiple experiments, it was found that using 256 hidden layers yielded the best results. Based on regularization theory [45] and experimental tuning, it was discovered that selecting 4 layers for the Transformer encoder can effectively prevent overfitting or underfitting, and achieve good convergence when the learning rate is set to 1e-4.

For the buoy's actual measurement dataset, due to the increase in time resolution from 1 hour to 10 minutes, the training strategy was adjusted accordingly: to ensure consistency in time span, the input sequence length was extended from 24 to 144; at the same time, as the time dimension included minute information, minute features were added to the input dimension; other parameters were also adjusted synchronously. The corresponding training strategy is detailed in Table 2.

**Table 1. Hyperparameter configuration of the ST-DAN for ERA5 dataset.**

| Model parameters | Value |
|---|---|
| Input sequence length | 24 |
| Step size | 1 |
| Predicted length | 1 |
| Input dimension | 7 |
| Hidden size | 256 |
| Attention heads | 8 |
| Transformer encoder layers | 4 |
| Learning rate | 1e-4 |
| Epochs | 100 |

**Table 2. Hyperparameter configuration of the ST-DAN for buoy observed data.**

| Model parameters | Value |
|---|---|
| Input sequence length | 144 |
| Step size | 1 |
| Predicted length | 1 |
| Input dimension | 8 |
| Hidden size | 128 |
| Attention heads | 4 |
| Transformer encoder layers | 3 |
| Learning rate | 5e-4 |
| Epochs | 150 |

## 4. Experiment and result analysis

This section focuses on the experiments and result analysis. Firstly, ablation experiments are designed to replace or remove various components of the model in order to verify the effectiveness of the temporal and spatial links in the proposed model. Secondly, comparative experiments are conducted by comparing the target elements (air temperature and wind speed) of two different datasets with mainstream temporal interpolation models, in order to verify the effectiveness of the *ST-DAN* model proposed in this paper. Finally, the *ST-DAN* model is trained using experimental datasets with good data quality and real-world datasets containing outliers, respectively, and the robustness of the model is verified through prediction results.

### 4.1. Ablation experiments

First, ablation studies were conducted based on the ERA5 dataset to validate the effectiveness of the proposed model's components from three dimensions: the temporal pathway, the spatial pathway, and the spatio-temporal dual pathway. For the temporal pathway, a comparison was made with a standalone Transformer model; for the spatial pathway, a comparison was made with the traditional Graph Attention Network (GAT). The experimental results are shown in Table 3, presenting evaluations for both the training and testing phases. The MAE, MSE, and RMSE values for the single temporal pathway and single spatial pathway network models were consistently worse than those of the spatio-temporal dual-pathway network model (ST-DAN).

Specifically, ST-DAN achieved an MAE of 0.0386, which is 37.6% and 18.2% lower than ST-GAT and the standalone Transformer model, respectively. Its MSE was 0.0262, representing reductions of 23.8% and 1.3% compared to ST-GAT and Transformer, respectively. The RMSE was 0.1617, showing reductions of 12% and 0.6% compared to ST-GAT and Transformer, respectively. These results validate the effectiveness of the individual model components and fully demonstrate the necessity of spatio-temporal dual-pathway fusion and its performance advantages.

### 4.2. Comparative experiments

To validate the effectiveness and superiority of the proposed ST-DAN model in outlier correction tasks, comparative experiments were conducted with the statistical model ARIMA, sequential models from the RNN family, models from the Transformer family, and the spatio-temporal dual-branch model ST-LSTM.

As a classical time series analysis tool, ARIMA achieves data stationarity through differencing and combines autoregressive and moving average processes to capture long-term trends and short-term fluctuations in the data. The RNN family includes RNN, Bi-GRU, and Bi-LSTM. These models leverage the inherent structure of hidden layer neurons to endow the model with memory capacity for sequential data, enabling them to learn temporal dependencies within the data. Bi-GRU and Bi-LSTM are bidirectional variants of GRU and LSTM, respectively, capable of learning contextual information from both past and future, effectively handling long sequence problems. The Transformer family includes Transformer, Informer, and Autoformer. These models utilize the attention mechanism to capture long-range temporal

**Table 3. Comparison of the performance of GAT, Transformer and ST-DAN in ablation experiments.**

| Model | MAE | MSE | RMSE | R2 |
|---|---|---|---|---|
| GAT-train | 0.0578 | 0.0495 | 0.2225 | 0.9943 |
| GAT-test | 0.0674 | 0.0586 | 0.2421 | 0.9850 |
| Transformer-train | 0.0462 | 0.0328 | 0.1811 | 0.9959 |
| Transformer-test | 0.0517 | 0.0458 | 0.2139 | 0.9832 |
| ST-DAN-train | **0.0243** | **0.0247** | **0.1571** | **0.9973** |
| ST-DAN-test | **0.0386** | **0.0262** | **0.1619** | **0.9996** |

dependencies for higher prediction reliability. Additionally, the ST-LSTM model was constructed by replacing the Transformer encoder in ST-DAN with an LSTM model, serving as a spatio-temporal dual-branch comparison model.

To ensure the fairness of the comparative experiments, the RNN family models (RNN, Bi-GRU, Bi-LSTM) used identical training parameter settings during training. Similarly, the Transformer family models (Transformer, Informer, Autoformer) maintained consistent internal parameters. To address the architectural differences between the two families, hyperparameter adjustments were made based on the proposed ST-DAN model's hyperparameters for key parameters such as learning rate, hidden layer dimension, and number of network layers. This aimed to make models from different families comparable in terms of architectural depth and overall capacity. Meanwhile, the number of training epochs and batch size remained consistent for all models to ensure uniform training conditions. Furthermore, the same early stopping threshold mechanism was applied across all experiments, effectively suppressing overfitting while fully exploring the performance potential of each model, thereby enabling a comparative evaluation of optimal performance under fair training conditions.

In the experiments, the eight models—ARIMA, RNN, Bi-GRU, Bi-LSTM, ST-LSTM, Informer, Autoformer, and ST-DAN—were used to perform missing value imputation and outlier correction on the temperature data from the ERA5 dataset. The performance metrics of the eight models are shown in Table 4, including both training and test set evaluations. The table shows that, except for the ARIMA model which does not involve a training/prediction split, all models experienced a performance drop in the testing phase compared to the training phase. Since the actual number of input and output variables for all models was 1, the values for $AR^2$ and $R^2$ are identical.

The proposed ST-DAN model demonstrated significant advantages across all four metrics: MAE, MSE, RMSE, and $R^2$. Its MAE was 0.0386, representing a 29.3% reduction compared to the relatively well-performing ST-LSTM model; MSE was 0.0262, a reduction of 54.9%; RMSE was 0.1619, a reduction of 32.9%; and $R^2$ was 0.9996, an improvement of 0.4%. These results indicate that the ST-DAN model better captures the underlying trends of the real data, and the corrected data possesses higher reference value.

Figs 8 and 9 provide a visual comparison using bar charts of the training evaluation, from which it is evident that the proposed ST-DAN model demonstrated the best performance across all metrics during the training phase.

**Table 4. Comparison of the performance of eight models in comparative experiments on the ERA5 dataset.**

| Model | MAE | MSE | RMSE | R2 | AR2 |
|---|---|---|---|---|---|
| ARIMA | 0.0683 | 0.1185 | 0.3442 | 0.9893 | 0.9893 |
| RNN-train | 0.0312 | 0.0668 | 0.2585 | 0.9878 | 0.9878 |
| RNN-test | 0.0581 | 0.1011 | 0.3176 | 0.8936 | 0.8936 |
| Bi-GRU-train | 0.0398 | 0.0726 | 0.2694 | 0.9911 | 0.9911 |
| Bi-GRU-test | 0.0576 | 0.0987 | 0.3142 | 0.9579 | 0.9579 |
| Bi-LSTM-train | 0.0431 | 0.0712 | 0.2668 | 0.9915 | 0.9915 |
| Bi-LSTM-test | 0.0564 | 0.0947 | 0.3077 | 0.9436 | 0.9436 |
| ST-LSTM-train | 0.0938 | 0.0703 | 0.2651 | 0.9928 | 0.9928 |
| ST-LSTM-test | 0.0538 | 0.0581 | 0.2410 | 0.9947 | 0.9947 |
| Informer-train | 0.0257 | 0.1093 | 0.3306 | 0.9785 | 0.9785 |
| Informer-test | 0.0514 | 0.1506 | 0.3881 | 0.9217 | 0.9217 |
| Autoformer-train | 0.0395 | 0.0597 | 0.2443 | 0.9926 | 0.9926 |
| Autoformer-test | 0.0543 | 0.1174 | 0.3426 | 0.9763 | 0.9763 |
| ST-DAN-train | **0.0243** | **0.0247** | **0.1571** | **0.9973** | **0.9973** |
| ST-DAN-test | **0.0386** | **0.0262** | **0.1619** | **0.9996** | **0.9996** |

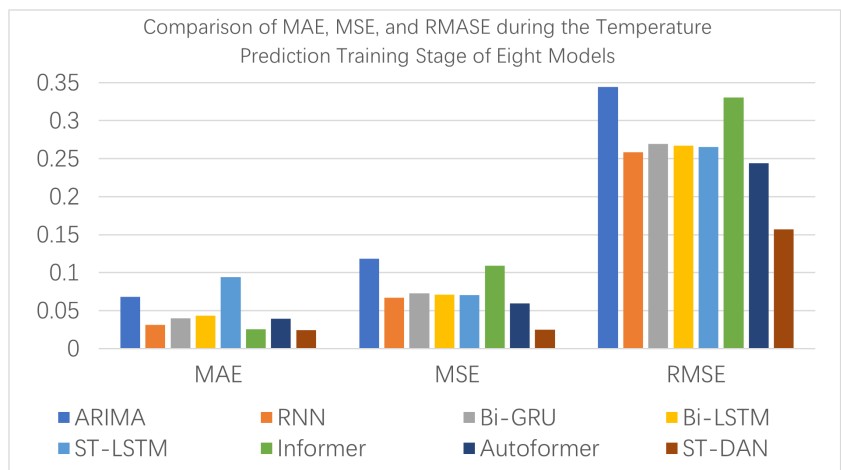

**Fig 8. Comparison of MAE, MSE, and RMASE during the Temperature Prediction Training Stage of Eight Models.**

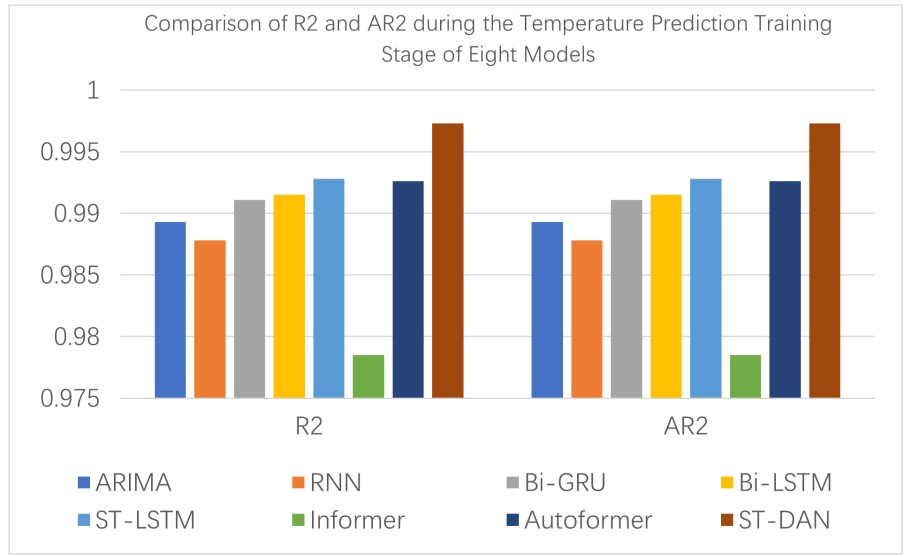

**Fig 9. Comparison of R2 and AR2 during the Temperature Prediction Training Stage of Eight Models.**

Figs 10 and 11 present intuitive comparative bar charts for the testing evaluation across various metrics. Compared to the training phase, all models exhibited a decline in their metrics to varying degrees; however, the ST-DAN model proposed in this paper still demonstrated the best performance during the testing process.

In the wind speed prediction experiment using buoy-measured data, a comparative analysis was conducted involving Bi-GRU, Bi-LSTM, Transformer, Informer, Autoformer, and the ST-DAN model. The experimental results for both the training and testing phases are presented in Table 5.

Compared to the experimental results on the ERA5 dataset (Table 4), in the context of buoy-measured data, the actual data exhibit stronger nonlinear fluctuation characteristics due to the complex and variable marine observation

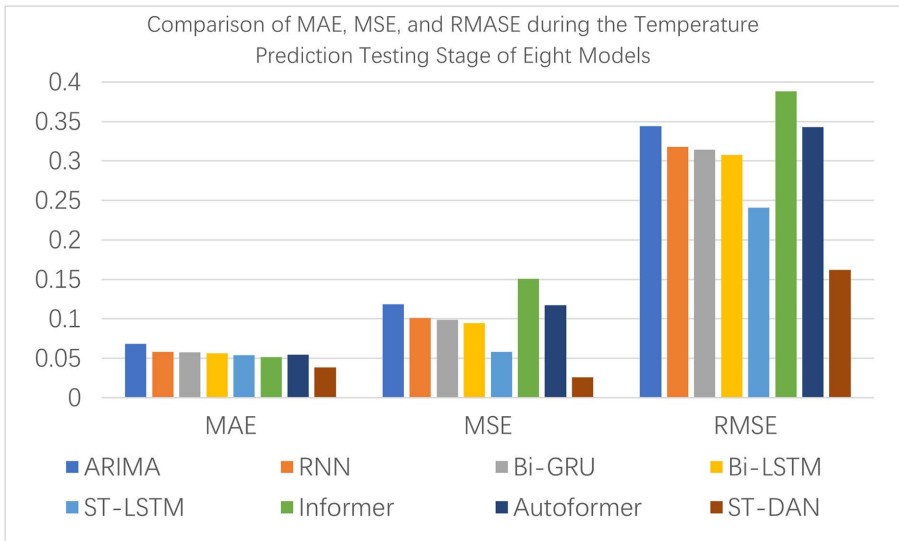

**Fig 10. Comparison of MAE, MSE, and RMASE during the Temperature Prediction Testing Stage of Eight Models.**

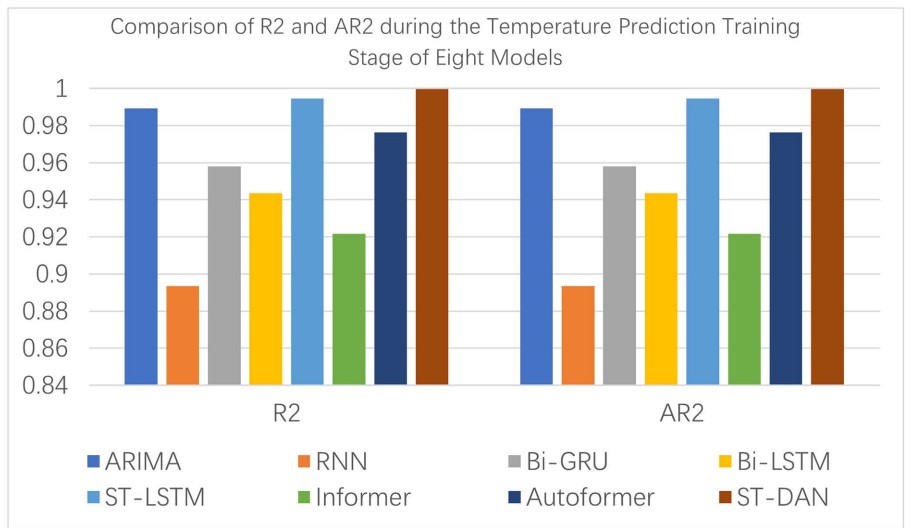

**Fig 11. Comparison of R2 and AR2 during the Temperature Prediction Testing Stage of Eight Models.**

environment. The inclusion of such fluctuating data in the training set led to a certain degree of decline in the overall prediction accuracy of all models. Nevertheless, the proposed ST-DAN model still maintained a high level of performance.

Among the compared models in the testing phase, the Transformer model demonstrated relatively good performance, with MAE, MSE, RMSE, $R^2$, and $AR^2$ values of 0.1543, 0.1697, 0.4119, 0.9028, and 0.9028, respectively. However, compared to Transformer, the ST-DAN model showed significant advantages across all metrics: an MAE of 0.0833, a reduction of 46.0%; an MSE of 0.1036, a reduction of 38.9%; an RMSE of 0.3218, a reduction of 21.9%; and an $R^2$ of 0.9732, an improvement of 7.2%.

**Table 5. Comparison of the performance of six models in comparative experiments on the buoy data.**

| Model | MAE | MSE | RMSE | R2 | AR2 |
|---|---|---|---|---|---|
| Bi-GRU-train | 0.1635 | 0.1051 | 0.3242 | 0.8995 | 0.8995 |
| Bi-GRU-test | 0.1791 | 0.1968 | 0.4436 | 0.8646 | 0.8646 |
| Bi-LSTM-train | 0.1313 | 0.1053 | 0.3246 | 0.8854 | 0.8854 |
| Bi-LSTM-test | 0.1695 | 0.1711 | 0.4136 | 0.8994 | 0.8994 |
| Transformer-train | 0.0829 | 0.1044 | 0.3231 | 0.9355 | 0.9355 |
| Transformer-test | 0.1543 | 0.1697 | 0.4119 | 0.9028 | 0.9028 |
| Informer-train | 0.0549 | 0.1655 | 0.4068 | 0.9035 | 0.9035 |
| Informer-test | 0.1895 | 0.2143 | 0.4629 | 0.8745 | 0.8745 |
| Autoformer-train | 0.0769 | 0.0873 | 0.2954 | 0.9247 | 0.9247 |
| Autoformer-test | 0.1879 | 0.1881 | 0.4337 | 0.8987 | 0.8987 |
| **ST-DAN-train** | **0.0542** | **0.0813** | **0.2851** | **0.9987** | **0.9987** |
| ST-DAN-test | **0.0833** | **0.1036** | **0.3218** | **0.9732** | **0.9732** |

It is noteworthy that, since the number of input and output variables was one for all models, the values of the adjusted coefficient of determination (AR²) and the coefficient of determination (R²) are identical. The experimental results validate the significant advantages of the ST-DAN model in wind speed prediction tasks under complex real-world marine environments.

Figs 12 and 13 present intuitive comparative bar charts of the evaluation data for each model during the training phase. Due to the presence of some latent outliers in the training set, the performance of all models declined to some extent; however, the proposed ST-DAN still remained optimal.

As shown in Figs 14 and 15, during the testing phase of wind speed prediction, the ST-DAN model exhibited only a minor performance difference compared to its training phase, indicating excellent generalization capability. Meanwhile, the

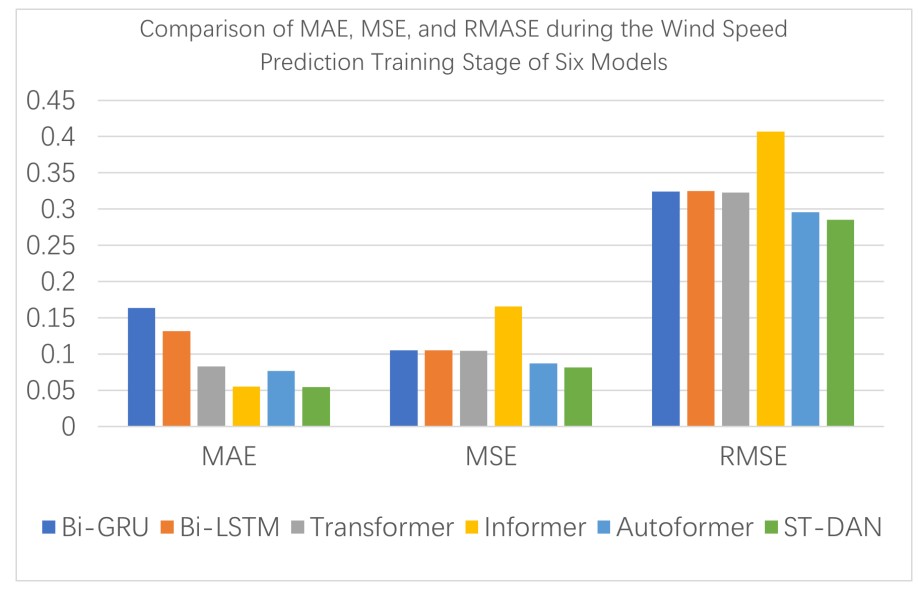

**Fig 12. Comparison of MAE, MSE, and RMASE during the Wind Speed Prediction Training Stage of Six Models.**

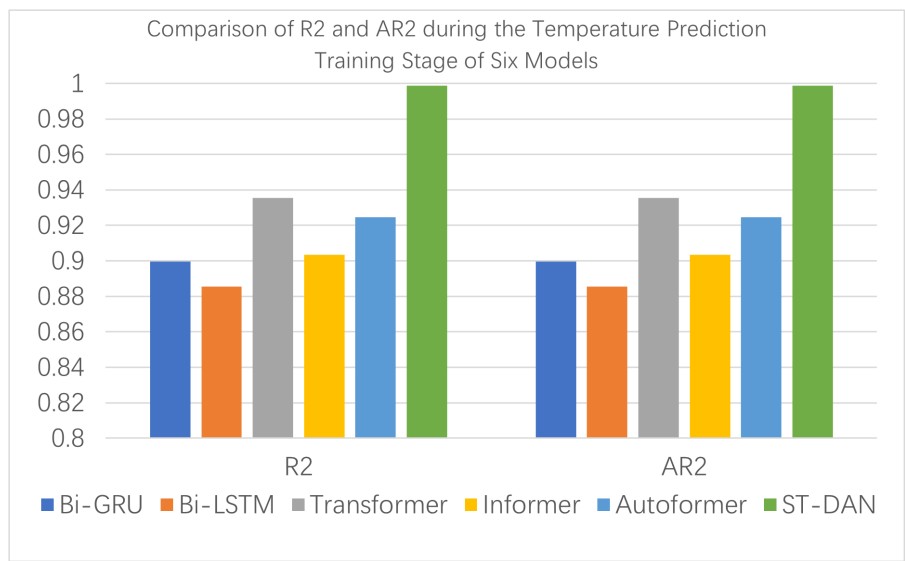

Fig 13. Comparison of R2 and AR2 during the Temperature Prediction Training Stage of Six Models.

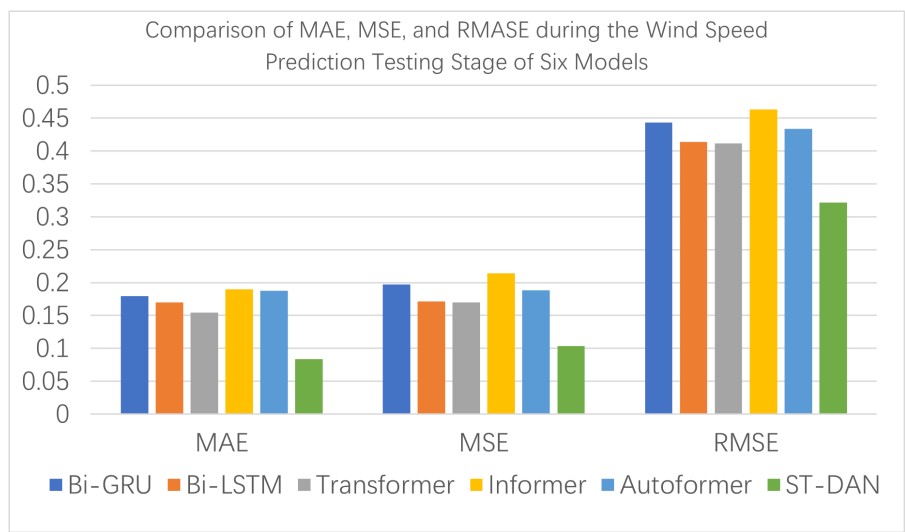

Fig 14. Comparison of MAE, MSE, and RMASE during the Wind Speed Prediction Testing Stage of Six Models.

model maintained its superior performance over all other compared models. These results collectively demonstrate that ST-DAN possesses reliable data restoration capability and strong robustness when handling complex meteorological data.

### 4.3. Visual analysis of data repair results

This study selects the relatively well-performing Transformer model and the ST-DAN model for comparative analysis of their effectiveness in temperature data restoration. The results before and after temperature data restoration are shown in Figs 16 and 17, where the solid blue line represents the true values, the red dashed line represents the predicted values, red circles indicate missing values, and black dots mark anomalies.

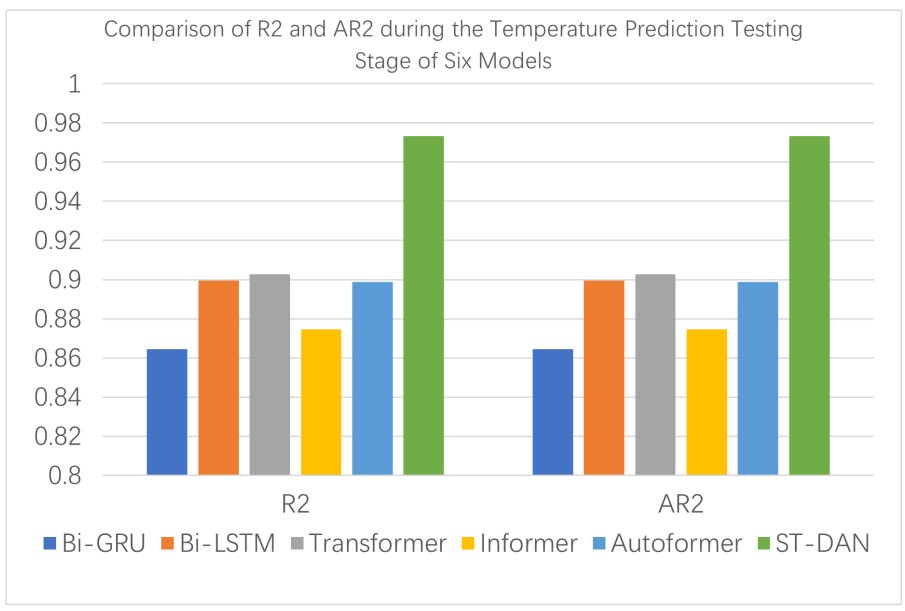

**Fig 15. Comparison of R2 and AR2 during the Temperature Prediction Testing Stage of Six Models.**

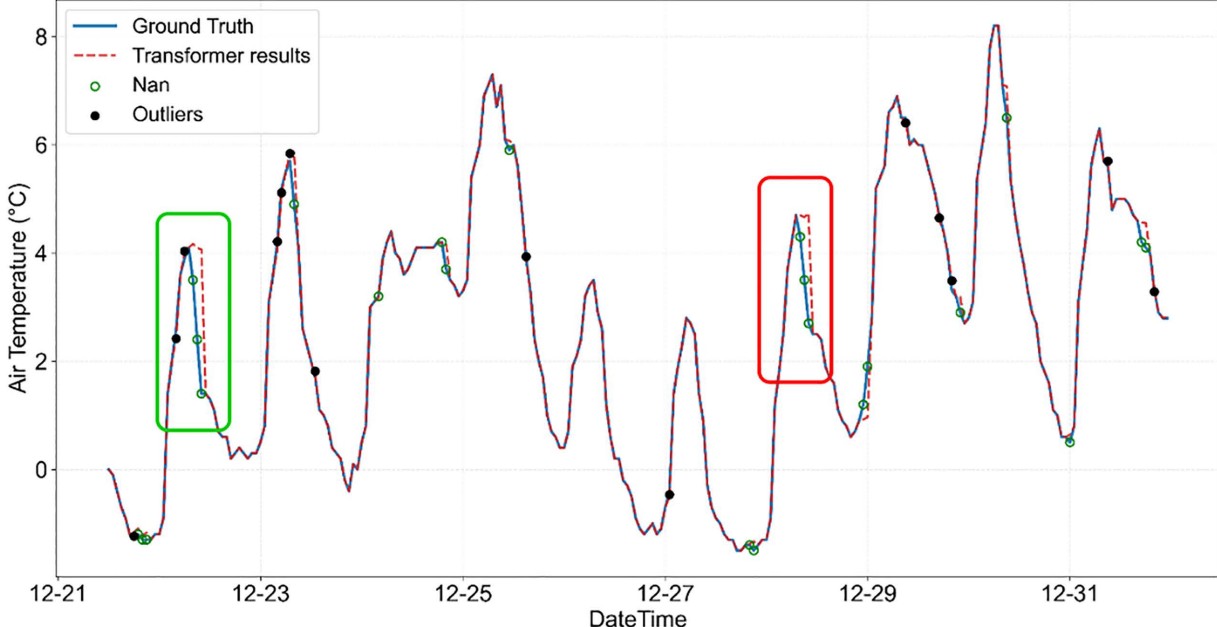

**Fig 16. The effect of temperature data repair based on Transformer.**

In Figs 16 and 17, the red and green boxes mark two regions with significant differences, whose enlarged local details are shown in Figs 18 and 19, respectively, covering a total of seven data points requiring restoration. Based on the discrepancy between the restored data (represented by the red dashed line) and the true data (represented by the solid blue

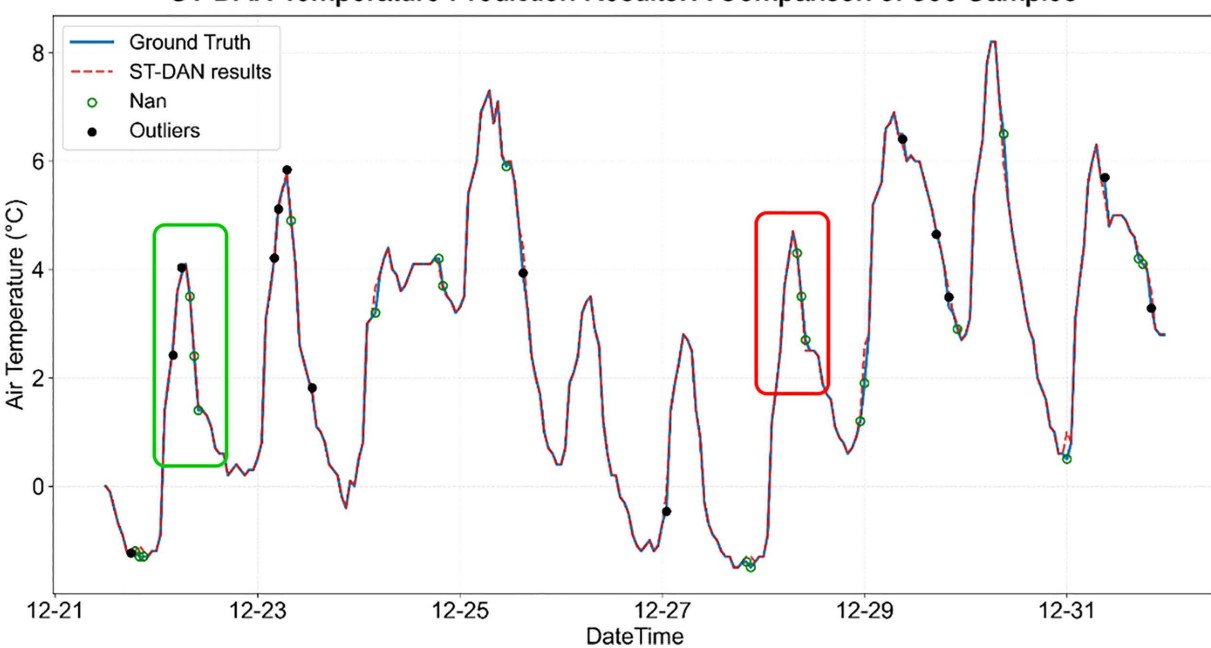

**Fig 17. The effect of temperature data repair based on ST-DAN.**

line), it can be concluded that the data restored by ST-DAN almost perfectly align with the true values, with no predicted values deviating from the underlying trend.

In Figs 16 and 17, the red and green boxes mark two regions with significant differences, whose enlarged local details are shown in Figs 18 and 19, respectively, covering a total of seven data points requiring restoration. Based on the discrepancy between the restored data (represented by the red dashed line) and the true data (represented by the solid blue line), it can be concluded that the data restored by ST-DAN almost perfectly align with the true values, with no predicted values deviating from the underlying trend (Table 6).

The results of the wind speed data before and after restoration are shown in Figs 20 and 21. Similarly, the solid blue line represents the ground truth values, the red dashed line denotes the predicted values, and the red hollow circles indicate missing values. Due to the presence of genuine anomalies in the measured data, the training set contained some anomalous samples, leading to a certain decline in overall prediction quality.

The results of the wind speed data before and after restoration are shown in Figs 20 and 21. Similarly, the solid blue line represents the ground truth values, the red dashed line denotes the predicted values, and the red hollow circles indicate missing values. Due to the presence of genuine anomalies in the measured data, the training set contained some anomalous samples, leading to a certain decline in overall prediction quality (Figs 22 and 23).

The ground truth values at these locations, along with the restoration results from the Transformer model and the ST-DAN model, are detailed in Table 7. The results demonstrate that despite interference from anomalous data, ST-DAN effectively captures the underlying variation trends of the data, exhibiting a favorable goodness of fit. This further validates the strong robustness of the proposed model.

### 4.4. Analysis of computational complexity

The variables in the computational complexity analysis are defined as follows: $B$ represents the batch size, $L$ denotes the length of the input sequence, $H$ signifies the dimension of the hidden layer, $n$ stands for the number of Transformer

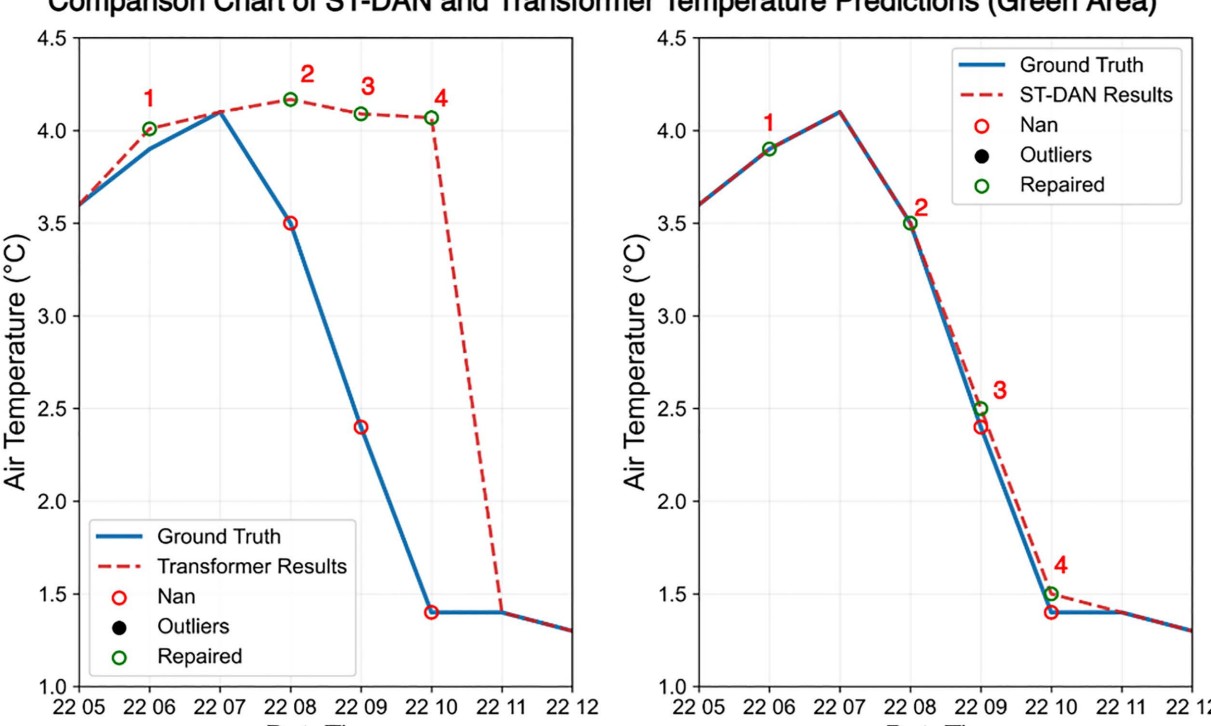

**Fig 18. Comparison of temperature data repair effects in the green area of Figs 16 and 17 (left: Transformer, right: ST-DAN).**

layers, and $C$ indicates the number of meteorological features. For time complexity, in the data preprocessing stage, time features are added to $n$ pieces of data and standardized, with a time complexity of $O(n)$. In the model computation part, the spatial encoding link involves processing the physical relationships and mask matrix calculations of 5-dimensional meteorological symbols, with a time complexity of $c$; in the temporal encoding link, the time complexity of the multi-layer Transformer encoder and its included multi-head attention mechanism is $O\left(n \times B \times L^2\right)$, while the feature fusion layer involves two linear transformation operations, with a time complexity of $O\left(B \times L \times H^2\right)$. Therefore, the total time complexity is $O\left(n \times B \times L^2 + B \times L \times H^2\right)$.

Regarding spatial complexity, in the temporal encoding link, it is primarily related to the number of Transformer layers, denoted as $n$, and is expressed as $O\left(n \times H^2\right)$. In the spatial encoding link, the spatial complexity is mainly determined by the physical relationship matrix and is expressed as $O\left(H^2\right)$. The remaining part involves the feature fusion layer, with spatial complexities of $O(C \times H)$ respectively. Therefore, the total spatial complexity is expressed as $O\left(n \times H^2 + B \times L \times H\right)$.

### 4.5. Robustness analysis

The *ST-DAN* model described in this paper leverages advanced concepts of spatio-temporal feature fusion and multi-head self-attention algorithms. It capitalizes on the technical advantages of its dual link architecture to establish a physical constraint mechanism suitable for complex anomaly scenarios in meteorological data repair, demonstrating strong robustness. The spatial link models the physical relationships between meteorological variables, while the temporal link captures the evolutionary patterns of target variables. This decoupled design prevents the cross-contamination of spatio-temporal features, ensuring that when one feature link is compromised by anomalies, the other can maintain baseline predictive

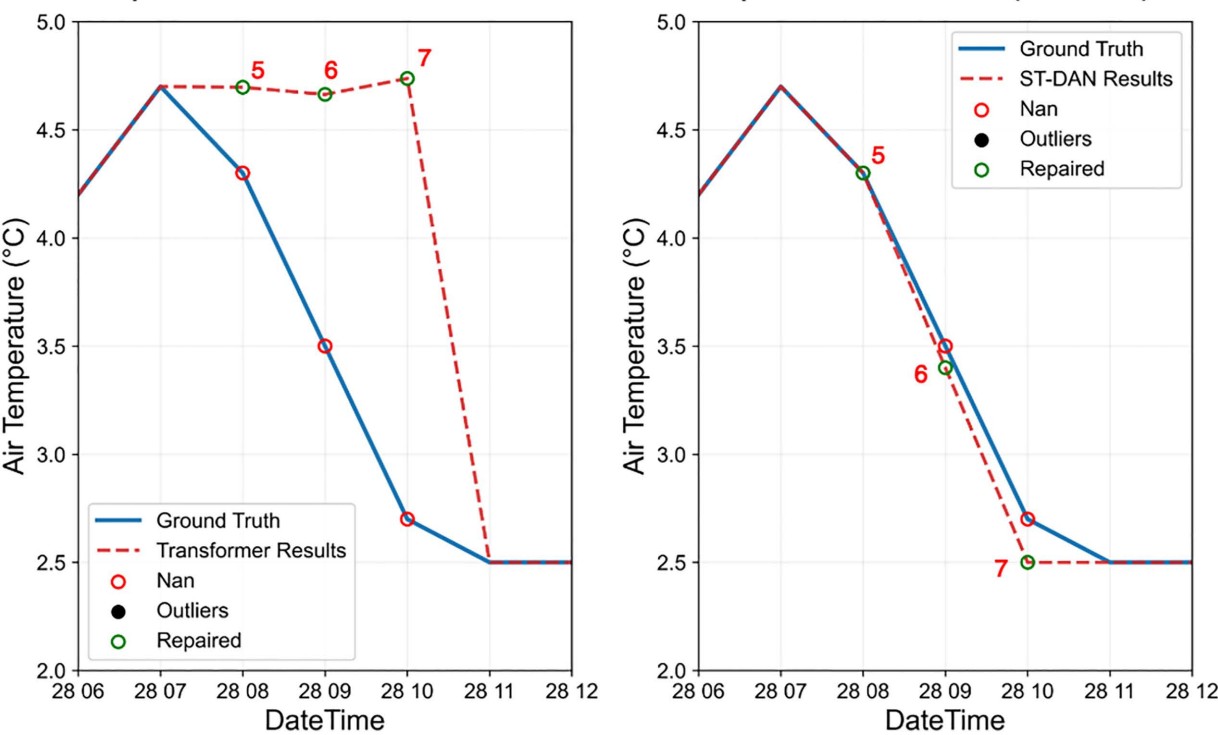

**Fig 19. Comparison of temperature data repair effects in the red area of Figs 16 and 17 (left: Transformer, right: ST-DAN).**

**Table 6. Comparison of specific temperature repaired values and truth (°C).**

| Index | Time | ST-DAN repaired values (°C) | Transformer repaired values (°C) | Outlier (°C) | Ground Truth (°C) |
|---|---|---|---|---|---|
| 1 | 2024.12.22 6:00 | 3.9 | 4.1 | 4.2 | 3.9 |
| 2 | 2024.12.22 8:00 | 3.5 | 4.2 | Nan | 3.5 |
| 3 | 2024.12.22 9:00 | 2.6 | 4.1 | Nan | 2.4 |
| 4 | 2024.12.22 10:00 | 1.6 | 4.1 | Nan | 1.4 |
| 5 | 2024.12.28 8:00 | 4.3 | 4.7 | Nan | 4.3 |
| 6 | 2024.12.28 9:00 | 3.5 | 4.6 | Nan | 3.5 |
| 7 | 2024.12.28 10:00 | 2.6 | 4.7 | Nan | 2.7 |

capability. Furthermore, by utilizing a physical constraint matrix based on learnable relationships derived from meteorological priors, the model is compelled to adhere to fundamental physical laws, thereby avoiding repaired results that contradict physical principles.

On the other hand, this paper adopts an unsupervised learning training scheme, using the training set of the *EAR5* numerical model for training and prediction. *ST-DAN* achieved high performance metrics, with *MAE* of 0.038, *MSE* of 0.0262, and *RMSE* of 0.1617. When subsequently trained using buoy observation data containing outliers, the *ST-DAN* model maintains high performance indicators, with *MAE* increasing by 30% to 0.0622, MSE increasing by 7.1% to 0.0487, and *RMSE* increasing by 3.7% to 0.2280. Although $R^2$ slightly decreases, it remains above 99% (see Table 7 for details).

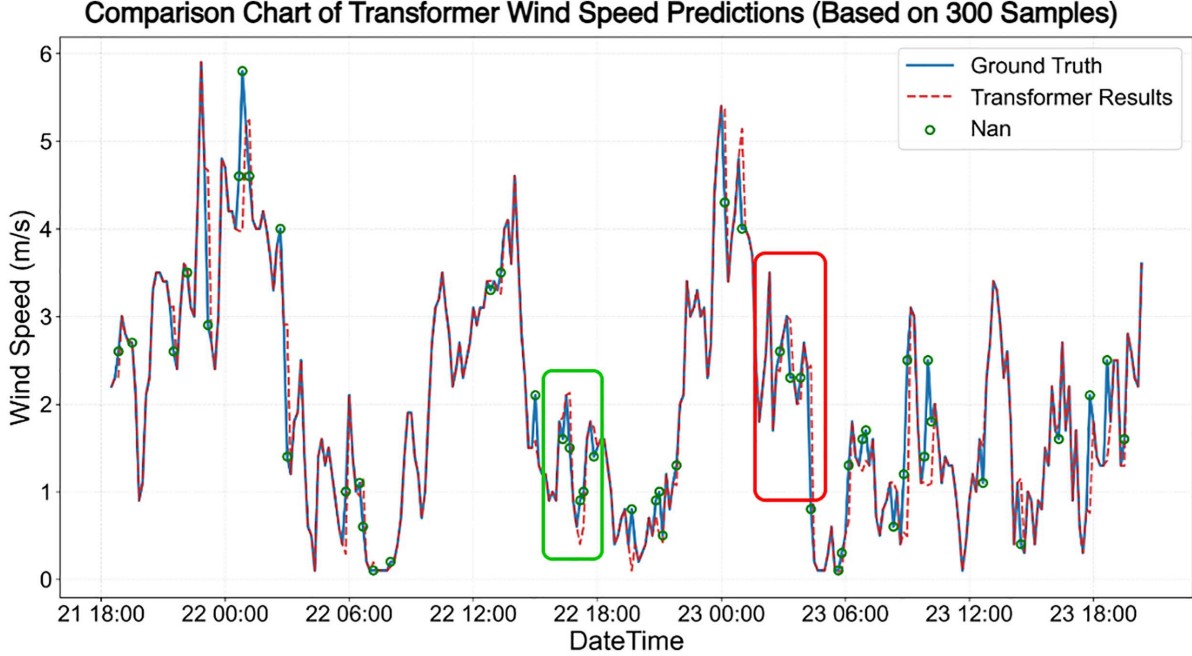

**Fig 20. The effect of wind speed data repair based on the Transformer model.**

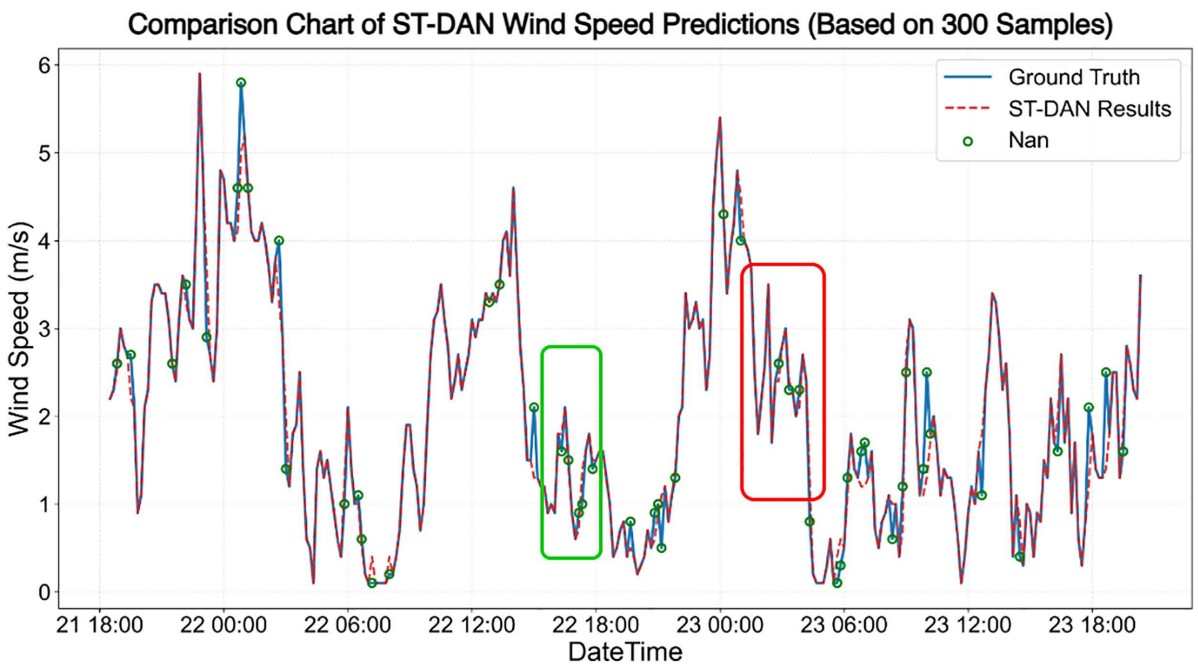

**Fig 21. The effect of wind speed data repair based on the ST-DAN model.**

**Fig 22. Comparison of repaired wind speed data in the green area of Figs 20 and 21 (left: Transformer, right: ST-DAN).**

Additionally, the data repair results for both *EAR5* numerical model data and buoy observation data do not violate the data variation trend. these experimental results collectively demonstrate the model's strong robustness.

## 5. Conclusion

The Spatio-Temporal Dual Attention Network *(ST-DAN)* proposed in this study integrates a Transformer encoder with an improved Graph Attention Network (GAT) to construct a dual-link reasoning architecture, effectively addressing the task of missing value imputation and outlier correction in buoy meteorological data. The temporal link employs the global self-attention mechanism of the Transformer encoder to accurately capture long-term temporal dependencies of individual elements, while the spatial link enhances the physical correlation between meteorological elements and dynamically optimizes weights by introducing a *GAT* model with a physically constrained adjacency matrix. The synergistic combination of the two significantly improves the accuracy and physical consistency of data repair. Experimental results show that *ST-DAN* performs excellently on both the *ERA5* dataset and actual buoy data from *Xiaomai* Island in *Qingdao*, and demonstrates good robustness in continuous missing scenarios and noisy data, validating its effectiveness in meteorological data repair tasks.

Nevertheless, the proposed model has certain limitations. The construction of the physical adjacency matrix relies on domain expertise, which limit its scalability; and the parallel computation of dual links results in slower training speed. Future research can be advanced in three aspects: first, exploring automated for generating constraint matrices by integrating data-driven approaches with physical rules to reduce dependence on expert knowledge; second, introducing sparse attention mechanisms and lightweight model design to optimize computational efficiency; third, extending the model to application such as multi-buoy collaborative data repair and high-frequency data imputation during extreme

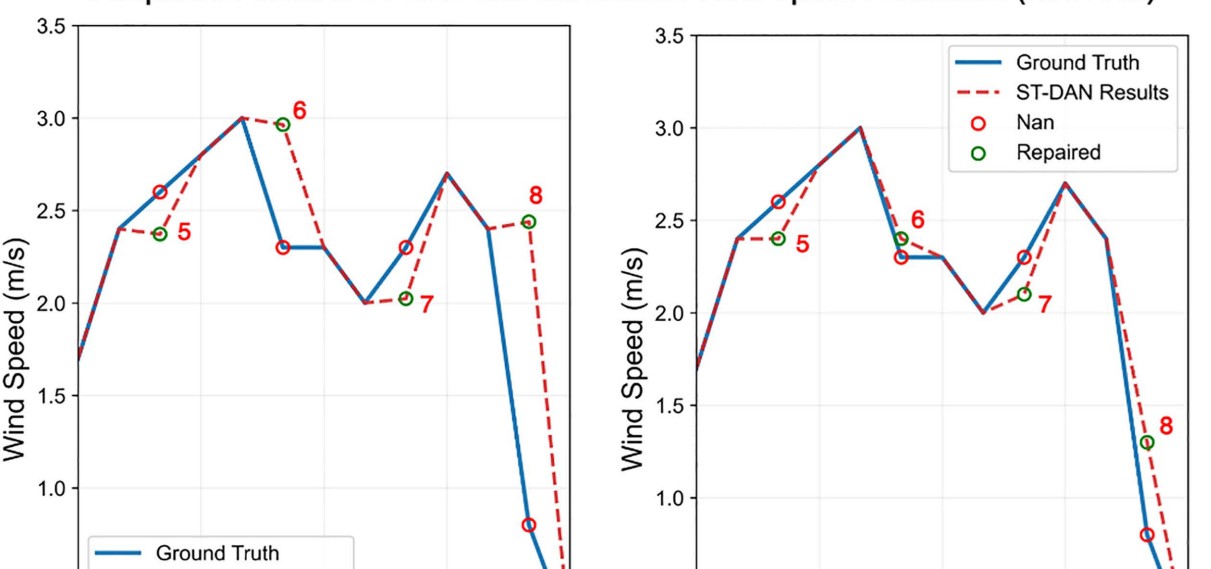

**Fig 23. Comparison of repaired wind speed data in the red area of Figs 20 and 21 (left: Transformer, right: ST-DAN).**

Table 7. Comparison of specific wind speed repaired values and truth.

| Index | Time | ST-DAN repaired values (m/s) | Transformer repaired values (m/s) | Ground Truth (m/s) |
|---|---|---|---|---|
| 1 | 2025-3-22 16:40 | 1.4 | 2.1 | 1.5 |
| 2 | 2025-3-22 17:10 | 0.7 | 0.4 | 0.9 |
| 3 | 2025-3-22 17:20 | 1.4 | 0.6 | 1 |
| 4 | 2025-3-22 17:50 | 1.5 | 1.7 | 1.4 |
| 5 | 2025-3-23 2:50 | 2.4 | 2.4 | 2.6 |
| 6 | 2025-3-23 3:20 | 2.4 | 2.9 | 2.3 |
| 7 | 2025-3-23 3:50 | 2.1 | 2 | 2.3 |
| 8 | 2025-3-23 4:20 | 1.3 | 2.4 | 0.8 |

weather events. Incorporating additional meteorological physical equations could further enhance the model's interpretability and generalization capability, thereby providing more comprehensive technical support for the high-quality application of marine meteorological observation data.

## Acknowledgments

The authors are grateful to the Ocean Buoy Technology Team from the Institute of Oceanographic Instrumentation, Shandong Academy of Sciences, for their invaluable assistance with the acquisition and processing of ocean buoy data.

## Author contributions

**Conceptualization:** Miaomiao Song, Shizhe Chen, Xiao Fu, Shixuan Liu.

**Data curation:** Jiuzhang Huang, Xiao Fu, Wenqing Li, Keke Zhang, Wei Hu.

**Formal analysis:** Miaomiao Song, Shizhe Chen, Wenqing Li.

**Funding acquisition:** Miaomiao Song, Xiao Fu, Shixuan Liu.

**Investigation:** Jiuzhang Huang, Shizhe Chen, Shixuan Liu, Keke Zhang.

**Methodology:** Miaomiao Song, Jiuzhang Huang, Shizhe Chen, Shixuan Liu, Keke Zhang.

**Project administration:** Shixuan Liu.

**Resources:** Xiao Fu, Shixuan Liu, Keke Zhang, Xingkui Yan.

**Software:** Jiuzhang Huang, Wenqing Li, Xingkui Yan.

**Supervision:** Miaomiao Song.

**Validation:** Jiuzhang Huang, Wei Hu, Xingkui Yan.

**Visualization:** Jiuzhang Huang, Wenqing Li, Wei Hu, Xingkui Yan.

**Writing – original draft:** Jiuzhang Huang.

**Writing – review & editing:** Miaomiao Song, Jiuzhang Huang.

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
