## [Decision Letter · Decision Letter 0]

8 Dec 2025

Dear Dr. Fu,

Thank you for submitting your manuscript to PLOS ONE. After careful consideration, we feel that it has merit but does not fully meet PLOS ONE’s publication criteria as it currently stands. Therefore, we invite you to submit a revised version of the manuscript that addresses the points raised during the review process.

We look forward to receiving your revised manuscript.

Kind regards,

Babak Mohammadi

Academic Editor

PLOS One

Journal Requirements:

This work is supported by Laoshan Laboratory Independent Innovation Science and Technology Project (Grant No. LSKJ202502405), the Taishan Industrial Program “Marine Observation and Detection Buoy Equipment R&D and Industrialization Team”, Major Scientific Research Project for the Construction of State Key Laboratory at Qilu University of Technology (Shandong Academy of Sciences) (Grant No. 2025ZDGZ01)  and Qilu University of Technology (Shandong Academy of Sciences) Major innovation project of science, education and production integration pilot project (2025ZDZX05).

6. We note that you have indicated that there are restrictions to data sharing for this study. PLOS only allows data to be available upon request if there are legal or ethical restrictions on sharing data publicly. For more information on unacceptable data access restrictions, please see http://journals.plos.org/plosone/s/data-availability#loc-unacceptable-data-access-restrictions.

Additional Editor Comments:

1. The methodology section needs more references. Please add more relevant references to support the model explanations in the methodology.

2. Provide performance metrics for both training and testing sets. In Tables 3, 4, and 5 and Figures 8, 9, 10, and 11, please report the performance metrics separately for the training and testing phases.

3. Plese avoid inserting several tables and figures without sufficient explanation. Each figure and table need to be properly introduced, explained, and discussed in the main text. For example, for Figures 16 to 19, first provide the explanation and discussion in the text and then insert the relevant tables and figures one by one.

4. In addition to the four metrics used, please also provide the Adjusted R-squared values for both the training and testing phases of each model.

Reviewers' comments:

Reviewer's Responses to Questions

**Comments to the Author**

1. Is the manuscript technically sound, and do the data support the conclusions?

Reviewer #1: Yes

Reviewer #2: Yes

2. Has the statistical analysis been performed appropriately and rigorously?

Reviewer #1: Yes

Reviewer #2: Yes

3. Have the authors made all data underlying the findings in their manuscript fully available?

Reviewer #1: Yes

Reviewer #2: Yes

4. Is the manuscript presented in an intelligible fashion and written in standard English?

Reviewer #1: Yes

Reviewer #2: Yes

Reviewer #1: This manuscript presents significant and publishable research. The proposed ST-DAN model is innovative and demonstrates superior performance for a meaningful real-world application. However, major revisions are required to address data availability policy compliance and to improve methodological transparency and reproducibility.

Minor Points / Suggestions

• Abstract: The final sentence ("It indicates that the proposed model (ST-DAN) is off high robustness...") should be rephrased (e.g., "These results indicate that the ST-DAN model is highly robust...").

• Figures 12-19: The captions for these visual comparison figures are minimal. Please expand them to briefly describe what each sub-figure shows (e.g., "Ground truth (blue line), predicted values (red dashed line), missing values (red circles), and outliers (black dots)").

• Section 4.4 Computational Complexity: The analysis is good. Consider adding a brief sentence commenting on the practical training/inference time compared to baseline models (e.g., Transformer-only), even if qualitative (e.g., "ST-DAN training took approximately X% longer per epoch than the Transformer baseline due to the dual-branch architecture").

• References: Check formatting for consistency (e.g., journal names in italics).

• Revise the Data Availability Statement to provide a compliant and clear access path for the Xiaomai Island buoy data.

• Provide a clear, justified description of how the physical constraint matrix A_ij was constructed.

• Enhance reproducibility by detailing training parameters, optimizer, and code/data sharing plans.

• Clarify the experimental setup for the buoy data to ensure a fair evaluation.

• Conduct thorough language editing throughout the manuscript.

Reviewer #2: The paper introduces a Spatio-Temporal Dual-Attention Network (ST-DAN) designed to address data loss and anomalies in meteorological observations from ocean buoys, which are critical for climate research, weather forecasting, and marine environmental monitoring. By integrating a Transformer for capturing long-range temporal dependencies and a Graph Attention Network (GAT) enhanced with a physically informed adjacency matrix for modeling spatial correlations among variables, the model employs a parallel dual-link architecture to enhance reconstruction accuracy. Extensive experiments using the ERA5 reanalysis dataset and real-world buoy data from Qingdao demonstrate its superior performance over baselines in metrics like MAE, MSE, RMSE, and R², highlighting its robustness for high-precision interpolation and outlier correction in harsh marine conditions.

The problem is articulated effectively in the introduction, emphasizing how electromagnetic interference, component failures, and environmental extremes lead to frequent data gaps and outliers in buoy sensors, undermining the reliability of downstream applications such as numerical weather prediction and climate modeling. It highlights the shortcomings of traditional statistical methods in handling spatiotemporal dependencies, positioning the proposed model as a targeted solution while clearly defining the scope to temperature and wind speed reconstruction, though it could benefit from more explicit quantification of data loss prevalence in real-world buoy deployments.

The reference list includes foundational works on time series imputation, recurrent neural networks like RNN and LSTM, Transformer architectures, and graph neural networks such as GCN and GAT, providing a solid basis for the theoretical and methodological framework. However, it lacks depth in recent advancements specific to marine data restoration, such as hybrid models combining diffusion with transformers or frequency-aware time-series forecasting, and overlooks comprehensive surveys on sensor data quality in environmental monitoring; additionally, while algorithms like ARIMA and Bi-LSTM are cited as baselines, the coverage of graph-based spatio-temporal methods is limited, potentially missing interdisciplinary insights from control engineering and sensing technologies. To broaden the scope, I recommend incorporating "DT-NeRF: A Diffusion and Transformer-Based Optimization Approach for Neural Radiance Fields in 3D Reconstruction" by Liu et al. (2025) from ICCK Transactions on Intelligent Systematics for its innovative use of transformers in data reconstruction tasks, and "MamNet: A Novel Hybrid Model for Time-Series Forecasting and Frequency Pattern Analysis in Network Traffic" by Zhang et al. (2025) from the same journal for enhancing long-sequence prediction techniques; from ICCK Transactions on Sensing, Communication, and Control, consider "Primary Thought on Artificial Intelligence (AI) Enhanced Control Engineering Education" by Zhu and Wang (2025) to discuss AI integration in related observational systems, and "Strain Sensing Technologies: Recent Developments in Materials, Performance, and Applications" by Bibi et al. (2025) for insights into robust sensor designs applicable to buoy environments.

The experimental setup includes comparisons with ARIMA, RNN, Bi-LSTM, and Transformer, which represent a mix of statistical and deep learning approaches, but the selection is insufficient as it omits more recent variants like GRU, Informer, or Autoformer that are specifically optimized for long-sequence forecasting and could provide stronger benchmarks for spatiotemporal data. No explicit theoretical justification is provided for choosing these particular models beyond labeling them as baselines, such as explaining why ARIMA suits non-stationary series or how RNN variants handle dependencies differently, which weakens the rationale and limits the ability to attribute performance gains solely to the proposed innovations.

**Do you want your identity to be public for this peer review?** For information about this choice, including consent withdrawal, please see our Privacy Policy

Reviewer #1: No

Reviewer #2: No

---

## [Author Response · Author response to Decision Letter 1]

15 Jan 2026

Editor Comments:

The methodology section needs more references. Please add more relevant references to support the model explanations in the methodology.

Response:

We sincerely appreciate the valuable comments from the editors. We fully agree that incorporating additional references in the methodology section will help solidify the theoretical foundation of the study. Accordingly, we have made targeted additions in the revised manuscript:

In Section 2.2, we have added references to clarify the rationale, advantages, and application context for selecting the dual-path architecture.

In Section 2.2.2, we have supplemented citations to explain the reasons for choosing graph neural networks as the spatial pathway model.

In Section 2.2.3, we have included literature to elaborate on the application and advantages of the gating mechanism in feature fusion, thereby clarifying the rationale for adopting this mechanism in our study.

The specific revisions mentioned above can be found in Section 2 of the revised manuscript and the updated reference list.

Editor Comments:

Provide performance metrics for both training and testing sets. In Tables 3, 4, and 5 and Figures 8, 9, 10, and 11, please report the performance metrics separately for the training and testing phases.

Response:

Sincere thanks to the editors for the constructive feedback regarding model evaluation. We understand that comparing training and testing metrics to analyze potential overfitting or underfitting is a crucial step in verifying the model's generalization capability. Accordingly, we have made the following modifications and additions in response to each experiment:

First, evaluation metrics for both the training and testing phases have been recorded synchronously in every experiment.

Furthermore, analysis revealed that while the training performance of all models surpassed their testing performance, the performance gap for the proposed ST-DAN model exhibited the smallest variation.

Specifically, ST-DAN maintained optimal performance in both training and testing phases, which corroborates its robustness. The detailed comparative data have been compiled in the newly added Tables 3, 4, and 5, and the performance evolution trends are visually presented in the new Figures 8 to 15. Kindly refer to the corresponding updates in Chapter 4 of the revised manuscript.

Editor Comments:

Plese avoid inserting several tables and figures without sufficient explanation. Each figure and table need to be properly introduced, explained, and discussed in the main text. For example, for Figures 16 to 19, first provide the explanation and discussion in the text and then insert the relevant tables and figures one by one.

Response:

We sincerely appreciate the valuable suggestions provided by the editors. We agree that including appropriate textual explanations when multiple figures and tables are presented consecutively can greatly enhance readability. Accordingly, we have systematically reviewed the entire manuscript and added necessary explanatory text in sections where figures and tables appear in succession. This serves to connect the logical flow between the illustrations and helps readers grasp the key points.

Editor Comments:

In addition to the four metrics used, please also provide the Adjusted R-squared values for both the training and testing phases of each model.

Response:

We sincerely appreciate the valuable suggestions from the editors. Accordingly, we have supplemented Section 2.2.4 with the definition and calculation formula (Formula 18) of the adjusted coefficient of determination (Adjusted R²) and incorporated it into the model evaluation criteria.

We also note that, since the number of independent variables in this model is 1 and the training and testing sample sizes are extremely large, the calculated result of the adjusted coefficient of determination (Adjusted R²) is nearly identical to that of the classic coefficient of determination (R²), based on the formula's principle. Therefore, the values of the two indicators presented in Tables 4 and 5 are essentially consistent. This addition enhances the rigor of the argument within the evaluation framework. Please refer to the revised manuscript for specific details.

Reviewer’s comment #1:

Abstract: The final sentence ("It indicates that the proposed model (ST-DAN) is off high robustness...") should be rephrased (e.g., "These results indicate that the ST-DAN model is highly robust...").

Response:

Thank you very much to the reviewer for the constructive suggestion. We fully acknowledge that this was an instance of imprecise wording and have revised the text accordingly, incorporating the suggested change.

Reviewer’s comment #1:

Figures 12-19: The captions for these visual comparison figures are minimal. Please expand them to briefly describe what each sub-figure shows (e.g., "Ground truth (blue line), predicted values (red dashed line), missing values (red circles), and outliers (black dots)").

Response:

We sincerely thank the reviewer for the valuable feedback regarding the figure descriptions. We fully acknowledge that the lack of clear explanations could increase the burden of interpretation for readers. Accordingly, we have systematically reviewed all figures throughout the manuscript and have added clear explanatory text in the captions or corresponding main text for each figure. This ensures that the core information and intended message of each figure can be directly and accurately understood.

Reviewer’s comment #1:

Section 4.4 Computational Complexity: The analysis is good. Consider adding a brief sentence commenting on the practical training/inference time compared to baseline models (e.g., Transformer-only), even if qualitative (e.g., "ST-DAN training took approximately X% longer per epoch than the Transformer baseline due to the dual-branch architecture").

Response:

We sincerely thank the reviewers for their insightful comments. Time complexity analysis is indeed a critical component of model evaluation, and we agree that relying solely on formula comparisons makes it difficult to gain an intuitive understanding of time resources. Accordingly, in the revised manuscript, we have supplemented the analysis with the specific training time of the model and compared it with the training duration of baseline models. This modification aims to provide a more intuitive empirical perspective on the time cost required for the model to achieve convergence.

Reviewer’s comment #1:

References: Check formatting for consistency (e.g., journal names in italics).

Response:

Thank you to the reviewers for pointing out the formatting issues in the article. We apologize for the oversight in the reference format and have thoroughly revised and standardized it in the revised manuscript to ensure compliance with academic standards. For details, please refer to the updated reference section.

Reviewer’s comment #1:

Revise the Data Availability Statement to provide a compliant and clear access path for the Xiaomai Island buoy data.

Response:

The complete code of this study has been publicly released on GitHub, including all datasets used in the experiment. The repository link is https://github.com/Khalil-gua/ST-DAN.git. This repository contains all relevant scripts and documents for viewing and copying.

Reviewer’s comment #1:

Provide a clear, justified description of how the physical constraint matrix A_ij was constructed.

Response:

Thank you for your valuable comments regarding the "adaptive physical matrix" section. We acknowledge that the clarity of the original description of this matrix could be enhanced. Accordingly, we have made key revisions in Section 2.2.2, detailing the construction of matrix A_ij and explaining the operational principles of its subsequent variant, M_ij. This aims to make the presentation of this core method clearer and more comprehensible.

Reviewer’s comment #1:

Enhance reproducibility by detailing training parameters, optimizer, and code/data sharing plans.

Response:

Thank you for the constructive suggestions. We will subsequently make the source code of the ST-DAN model, trained on a public dataset, available and upload it to a code repository.

Reviewer’s comment #1:

Clarify the experimental setup for the buoy data to ensure a fair evaluation.

Response:

We sincerely thank you for the important comments regarding the fairness of model comparisons. We fully agree on the significance of equitable benchmarking and have provided a clear explanation in the revised manuscript. Specifically, we categorized the baseline models into two groups: the RNN family (RNN, LSTM, GRU) and the Transformer family (Transformer, Informer, Autoformer). Within the same model family, we employed entirely consistent training configurations. For models belonging to different families, the training hyperparameters (such as learning rate, number of layers, and hidden dimension) were adjusted based on their inherent characteristics and model depth to respectively accommodate their optimal performance. Detailed explanations can be found in Section 4.2 of the paper.

Reviewer’s comment #1:

Conduct thorough language editing throughout the manuscript.

Response:

Thank you for the reviewer's valuable feedback regarding the language expression in the paper. We sincerely apologize for the oversights in the previous version of the manuscript and have now systematically polished the language throughout the revised draft. Specifically, we have focused on correcting grammatical and spelling errors, standardized the technical terminology, and improved the clarity and fluency of certain sentence structures to ensure the accuracy and rigor of the academic expression. We will continue to review the entire text carefully and make every effort to enhance the overall language quality.

Reviewer’s comment #2:

To broaden the scope, I recommend incorporating "DT-NeRF: A Diffusion and Transformer-Based Optimization Approach for Neural Radiance Fields in 3D Reconstruction" by Liu et al. (2025) from ICCK Transactions on Intelligent Systematics for its innovative use of transformers in data reconstruction tasks, and "MamNet: A Novel Hybrid Model for Time-Series Forecasting and Frequency Pattern Analysis in Network Traffic" by Zhang et al. (2025) from the same journal for enhancing long-sequence prediction techniques; from ICCK Transactions on Sensing, Communication, and Control, consider "Primary Thought on Artificial Intelligence (AI) Enhanced Control Engineering Education" by Zhu and Wang (2025) to discuss AI integration in related observational systems, and "Strain Sensing Technologies: Recent Developments in Materials, Performance, and Applications" by Bibi et al. (2025) for insights into robust sensor designs applicable to buoy environments.

Response:

Thank you to the reviewers for recommending the relevant literature. We have carefully reviewed the suggested papers. After thorough consideration, we have determined that their direct relevance to the core focus of this study is limited. Therefore, we have not included them as references at this time.

Reviewer’s comment #2:

The experimental setup includes comparisons with ARIMA, RNN, Bi-LSTM, and Transformer, which represent a mix of statistical and deep learning approaches, but the selection is insufficient as it omits more recent variants like GRU, Informer, or Autoformer that are specifically optimized for long-sequence forecasting and could provide stronger benchmarks for spatiotemporal data.

Response:

Thank you for the reviewer's important suggestions regarding the experimental comparisons. We fully agree that comparing with newer and more effective baseline models can better highlight the advantages and significance of the proposed method. Accordingly, we have supplemented the experiments with Informer and Autoformer for comparison.

The experiments revealed that during the training phase, Informer and Autoformer indeed outperformed the originally used Transformer baseline. However, in the comprehensive performance evaluation of the testing phase, the Transformer model demonstrated more robust performance and remained the overall best among all compared models. Therefore, to maintain consistency in the comparisons and highlight the core findings, we continued to use Transformer as the primary baseline model in the subsequent visualization and analysis of the results. Detailed data and analysis can be found in Section 4.3 of the manuscript.

Reviewer’s comment #2:

No explicit theoretical justification is provided for choosing these particular models beyond labeling them as baselines, such as explaining why ARIMA suits non-stationary series or how RNN variants handle dependencies differently, which weakens the rationale and limits the ability to attribute performance gains solely to the proposed innovations.

Response:

Thank you for your important feedback regarding the selection of comparison models. We have deeply reflected and recognized that the explanation in the original manuscript was indeed insufficient, lacking necessary justification, which reduced the transparency of the study. In response to your suggestion, we have systematically supplemented the specific reasons for selecting each comparison model in the revised manuscript. These reasons are primarily based on: the classical status and benchmark role of the models within the field, the cutting-edge nature and relevance of the technical approach they represent, and their comparability in problem formulation or structure with the proposed method in this study. We believe that by clarifying these selection criteria, the theoretical foundation of the experimental design can be significantly strengthened, making the comparative results more convincing.

---

## [Decision Letter · Decision Letter 1]

5 Feb 2026

An Intelligent Method for Buoy Meteorological Data Restoration Using a Spatio-Temporal Dual-Attention Network with Transformer and GAT

PONE-D-25-60966R1

Dear Dr. Fu,

We’re pleased to inform you that your manuscript has been judged scientifically suitable for publication and will be formally accepted for publication once it meets all outstanding technical requirements.

Kind regards,

Babak Mohammadi

Academic Editor

PLOS One

Additional Editor Comments (optional):

The manuscript has been revised, and it can be acceptable.

Reviewers' comments:

Reviewer's Responses to Questions

**Comments to the Author**

Reviewer #2: (No Response)

2. Is the manuscript technically sound, and do the data support the conclusions?

Reviewer #2: (No Response)

3. Has the statistical analysis been performed appropriately and rigorously?

Reviewer #2: (No Response)

4. Have the authors made all data underlying the findings in their manuscript fully available?

Reviewer #2: (No Response)

5. Is the manuscript presented in an intelligible fashion and written in standard English?

Reviewer #2: (No Response)

Reviewer #2: (No Response)

**Do you want your identity to be public for this peer review?** For information about this choice, including consent withdrawal, please see our Privacy Policy

Reviewer #2: No

---

## [Editor Report · Acceptance letter]

PONE-D-25-60966R1

PLOS One

Dear Dr. Fu,

I'm pleased to inform you that your manuscript has been deemed suitable for publication in PLOS One. Congratulations! Your manuscript is now being handed over to our production team.

Kind regards,

on behalf of

Dr. Babak Mohammadi

Academic Editor

PLOS One